# REMSA: AN LLM AGENT FOR FOUNDATION MODEL SELECTION IN REMOTE SENSING

## ABSTRACT

Foundation Models (FMs) are increasingly integrated into remote sensing (RS) pipelines for applications such as environmental monitoring, disaster assessment, and land-use mapping. These models include unimodal vision encoders trained in a single data modality and multimodal architectures trained in multiple sensor modalities, such as synthetic aperture radar (SAR), multispectral, and hyperspectral imagery, or jointly in image-text pairs in vision-language settings. FMs are adapted to diverse tasks, such as semantic segmentation, image classification, change detection, and visual question answering, depending on their pretraining objectives and architectural design. However, selecting the most suitable remote sensing foundation model (RSFM) for a specific task remains challenging due to scattered documentation, heterogeneous formats, and complex deployment constraints. To address this, we first introduce the RSFM Database (**RS-FMD**), the first structured and schema-guided resource covering over 150 RSFMs trained using various data modalities, associated with different spatial, spectral, and temporal resolutions, considering different learning paradigms. Built on top of **RS-FMD**, we further present **REMSA** (**Re**mote-sensing **M**odel **S**election **A**gent), the first LLM agent for automated RSFM selection from natural language queries. **REMSA** combines structured FM metadata retrieval with a task-driven agentic workflow. In detail, it interprets user input, clarifies missing constraints, ranks models via in-context learning, and provides transparent justifications. Our system supports various RS tasks and data modalities, enabling personalized, reproducible, and efficient FM selection. To evaluate **REMSA**, we introduce a benchmark of 75 expert-verified RS query scenarios, resulting in **900** task-system-model configurations under a novel expert-centered evaluation protocol. **REMSA** outperforms multiple baselines, including naive agent, dense retrieval, and unstructured retrieval augmented generation based LLMs, showing its utility in real decision-making applications. **REMSA** operates entirely on publicly available metadata of open source RSFMs, without accessing private or sensitive data. Our code and data will be publicly released.

## 1 INTRODUCTION

With the growing availability of remote sensing (RS) missions and their onboard sensors (e.g., Sentinel-2 (Drusch et al., 2012), Sentinel-1 (Torres et al., 2012), EnMAP (Guanter et al., 2015)), RS plays an increasingly important role in many applications such as agriculture, disaster response, urban development, and biodiversity monitoring. These applications increasingly rely on foundation models (FMs) that can generalize across various RS data modalities with different spatial, spectral and temporal resolutions, geospatial extents and applications, while being transferable and effective even with limited labeled data. Recently, numerous FMs have emerged in the RS domain, offering powerful capabilities for interpreting complex RS data. These models include vision-only encoders trained on single or multiple RS data modalities (e.g., SatMAE (Cong et al., 2022), CROMA (Fuller et al., 2023)) and vision–language models (VLMs) trained

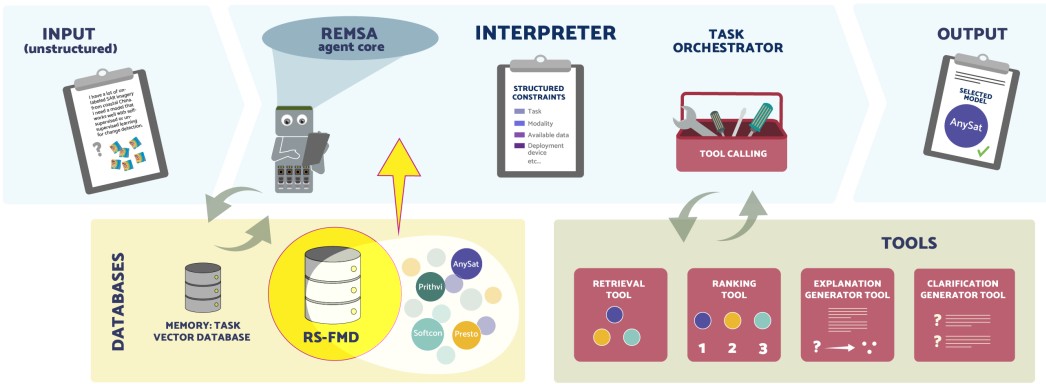

Figure 1: Framework of **REMSA**

jointly on RS data modalities and text (e.g., GRAFT (Mall et al., 2024), TEOChat (Irvin et al., 2025), Earth-Dial (Soni et al., 2025)). These models are pretrained on large-scale RS datasets encompassing a diverse range of sensor modalities, including RGB, multispectral, hyperspectral, synthetic aperture radar (SAR), and light detection and ranging (LiDAR), across multiple spatial and temporal resolutions. Each FM exhibits its strengths in distinct applications, such as classification, object detection, change detection, captioning, and visual question answering (VQA). For instance, in practice, change detection typically relies on multitem-poral SAR or optical data inputs, while fine-grained land cover mapping often benefits from high-resolution optical imagery. This diversity brings new possibilities for multi-modal RS applications, but it also raises the challenge of selecting the most suitable FM for a given task with data modality and operational constraints.

Despite these advances, selecting an FM that is suitable for a specific RS task remains a challenge. Users must balance diverse constraints such as the available data modalities and volume, geographic coverage, computational resources, and task-specific evaluation priorities. These constraints have been shown to significantly influence RSFM generalization and robustness (Purohit et al., 2025; Plekhanova et al., 2025). With hundreds of remote sensing foundation models (RSFMs) now publicly available (Guo et al., 2024; Li et al., 2025) and no unified structured schema to organize their properties (such as model architectures, training data, or reported performance), the selection process is often manual, time-consuming, and error-prone. Existing approaches rely on searching across scattered repositories and publications, manually parsing papers and model cards, and running exhaustive experiments (Ramachandran et al., 2025; Adorni et al., 2025), all without guaranteed reproducibility or transparency. Even public RS benchmarks (Lacoste et al., 2023; Simumba & et al.; Li et al., 2024) mainly compare model accuracy on fixed applications, offering little support for matching models to user-specific constraints or deployment trade-offs. This makes a unified, machine-readable database (DB) of RSFMs a necessary basis for any systematic selection and automation.

Recent advances in large language model (LLM) agent have shown the feasibility of combining language understanding, tool invocation, multi-turn interaction, and automatic structured reasoning to assist decision-making processes (Singh et al., 2024; Xiong et al., 2025; Agashe et al., 2025; Liu et al., 2025a). However, most LLM agents target general-purpose question answering. To our knowledge, no prior work has developed a domain-specific agent for FM selection in operational, constraint-heavy RS scenarios. In particular, RS tasks involve complex trade-offs across sensors, spatial, spectral, and temporal resolutions, as well as data availability. Existing LLMs lack the domain knowledge and structured access to model documentation to address these constraints. Hence, such an agent must provide more robust and interpretable solutions.

In this work, we first introduce the Remote Sensing Foundation Model Database (**RS-FMD**), the first schema-guided catalog of more than 150 RSFMs, covering various data modalities, pretraining strategies, and benchmark results. On top of **RS-FMD**, we propose **REMSA**, the first LLM-based agent for automated FM selection in RS. As shown in Figure 1, **REMSA** is a modular agent that automates FM selection through structured query interpretation and dynamic tool use. It extracts user intent from free text input and converts it into constraints. And based on the task state, the agent selectively calls tools to retrieve relevant FMs from **RS-FMD**, rank FMs using LLM-based reasoning, interact with the user in clarification loops, and provide transparent explanations. A memory mechanism further enhances accuracy and personalization. To evaluate **REMSA**, we build the first benchmark dataset of real user queries and establish an expert-driven evaluation protocol. We also implement a set of carefully constructed baselines, ensuring fair and meaningful comparisons with **REMSA**. **REMSA** is designed to support a broad range of end-users, including RS scientists, machine learning researchers, and industry practitioners who need to identify suitable RSFMs for their tasks. Because **REMSA** accepts free-text queries and incorporates structured interpretation together with multi-turn clarification, it can guide even non-experts who may not be familiar with RS modalities or FM architectures. This makes **REMSA** suitable for both exploratory use by practitioners and rigorous FM selection in research settings. Although **REMSA** uses a modular agent design, our contribution is methodological. We treat RSFM selection as a research problem of how FMs should be compared, selected, and deployed under real constraints. In summary, we make the following **contributions**:

- We introduce **RS-FMD**, the first structured and schema-guided DB of over 150 RSFMs. We will release it as a community resource with continuous maintenance and updates.
- We propose **REMSA**, a modular LLM agent that combines structured metadata grounding, dense retrieval, in-context ranking, clarification, explanation, memory augmentation, and a task-aware orchestration mechanism to support complex FM selection in real RS settings.
- We construct the **first** benchmark dataset and design an evaluation protocol for FM selection, encompassing 75 realistic queries across various RS tasks and provide 900 evaluation results.

## 2 RELATED WORK

**Foundation Models and Model Selection.** Due to the rapid emergence of RSFMs, there has been extensive research into their capabilities and benchmarks (Liu et al., 2025b; Wu et al., 2024; Pathak et al., 2025). In RS, recent surveys and benchmarks (Xiao et al., 2024; Ramachandran et al., 2025; Li et al., 2024) have systematically cataloged FMs and evaluated their performance on applications such as land cover classification, wildfire scar segmentation, urban change detection, visual question answering, etc. However, these works primarily focus on descriptive analysis or standardized evaluation, offering limited support for automated FM selection. Large-scale evaluations such as GEO-Bench-2 (Simumba & et al.) further highlight that RSFM performance varies strongly across capability dimensions, but still do not address automatic FM selection. Recent work also shows that pre-training data coverage (geographic and sensor diversity) strongly affects RSFM generalization (Purohit et al., 2025; Plekhanova et al., 2025). While current benchmarks document these properties, they do not use them to guide model choice, further motivating automated FM selection. Additionally, there is a new capabilities encoding approach that estimates a model's performance on unseen downstream tasks, reducing the need for exhaustive fine-tuning (Adorni et al., 2025). Although this provides valuable tools for comparative evaluation, it is still a benchmarking tool that does not address end-to-end automatic FM selection workflows. Moreover, previous surveys and benchmarks are static and task-specific, lacking a unified schema or machine-readable representation of RSFMs. In contrast, our **RS-FMD** consolidates the available FMs into a structured, extensible resource that directly supports automated retrieval, comparison, and selection. Another relevant line of work is AutoML, which includes frameworks such as Auto-WEKA (Thornton et al., 2013), Auto-sklearn (Feurer et al., 2015), and CAML (Neutatz et al., 2024). They automate the selection of parameters, algorithms, or pipelines through meta-learning and optimization techniques. Although these approaches show the feasibility of automating model choice in classical machine

learning settings, they have not been extended to the selection of FMs, particularly in the RS domain. To our knowledge, there is no existing autonomous method or agent that assists scientists in selecting the most suitable FM for their specific constraints and applications. Our work fills this gap by combining **RS-FMD** and **REMSA**, presenting the first domain-specialized agentic workflow for FM selection that automates the matching of user constraints to appropriate models.

**Tool-Augmented Agents in Remote Sensing.** Recent developments in retrieval-augmented language models and tool-augmented agents such as ReAct (Yao et al., 2023), HuggingGPT (Shen et al., 2023), and ToRA (Gou et al., 2024) show the feasibility of combining LLMs with structured retrieval and external tool invocation for complex reasoning and planning. In RS, several works have explored modular agentic workflows. GeoLLM-Squad (Lee et al., 2025) introduces a multi-agent orchestration framework that decomposes geospatial tasks into specialized sub-agents, improving scalability and correctness over single-agent baselines. RS-Agent (Xu et al., 2024) integrates retrieval pipelines and tool scheduling to process spatial question answering tasks, while ThinkGeo (Shabbir et al., 2025) introduces a benchmark for evaluating multi-step tool-augmented agents on RS workflows. Recently, TEOChat (Irvin et al., 2025) extended large vision-language assistants to temporal RS data by training on instruction following datasets, supporting conversational analysis of time-series data. These agents highlight the benefits of agent-based modularity and retrieval-augmented reasoning. However, they primarily target geospatial information extraction, change detection, or VQA applications rather than FM selection workflows. Our agent explicitly integrates a curated FM database with structured retrieval, agentic ranking, interactive constraint resolution, and transparent model reasoning, making it the first tool-augmented agent tailored for FM selection in RS.

## 3 REMOTE SENSING FOUNDATION MODEL DATABASE (**RS-FMD**)

**RS-FMD** is a curated DB of all RSFMs we could find (~150 RSFMs), serving as the structured knowledge base behind **REMSA**. It enables interpretable and constraint-aware FM selection by consolidating heterogeneous knowledge resources into a unified, machine-readable format. To build **RS-FMD**, we conducted a systematic search for RSFMs using multiple sources. We reviewed survey papers and popular FM lists, surveyed recent RS and ML venues, ran keyword searches on arXiv, and inspected linked GitHub repositories.

**Schema Design.** Each record follows a schema covering properties such as identifiers, architecture, modalities, and pretrained model weights, along with structured fields for pretraining datasets and benchmark evaluations. This schema ensures traceability, comparability, and extensibility across FMs. The full schema and an example record are in Appendix A.. This comprehensive schema enables our FM selection agent to ground its reasoning in model capabilities and match models to user-defined applications and constraints. It also ensures that critical properties, such as input data modalities, spatial, spectral, and temporal characteristics, and training configurations, can be queried and filtered in a principled and automated manner.

**Automated database population.** Populating this database requires extracting structured information from diverse sources, such as papers, model cards, and repositories. Due to the scale and heterogeneity of available model documentation, fully manual curation is impractical. Therefore, we adopt an automated knowledge extraction approach coupled with confidence-guided human verification. Our approach is a schema-guided LLM extraction pipeline inspired by a general knowledge extraction approach OneKE (Luo et al., 2025), but significantly adapted to our domain and use case. Specifically, we extend their approach by introducing our own schema definitions, adding a dedicated confidence scoring step, and optimizing prompt design for RS model descriptions. The process is entirely automated and iterative: for each FM, we collect and input a set of unstructured sources, then issue multiple LLM calls to generate independent structured outputs in each iteration. Each output is validated against the schema, parsed, and aggregated. This iterative strategy allows us to exploit both the probabilistic uncertainty of each iteration and the self-consistency across iterations. Fields for which the model produces divergent outputs or low log-probabilities are marked as uncertain and

passed to the human verification stage. The resulting pipeline effectively converts complex heterogeneous text sources into machine-readable JSON records with minimal manual intervention.

**Confidence Score for Human Verification.** Ensuring the reliability of the extracted metadata is critical for FM selection. To this end, we define a confidence score for each field in each record, enabling targeted human verification only where the uncertainty is high. Our confidence score combines two complementary criteria: the model's generation probability and the consistency of outputs across multiple LLM sampling rounds. For each field, we compute the confidence score as follows:

$$\text{Confidence} = w_{\text{logp}} \cdot \text{NormalizedLogProb} + w_{\text{cons}} \cdot \text{SelfConsistency} \tag{1}$$

where **NormalizedLogProb** quantifies the LLM's internal certainty by mapping the raw log-probability of the generated field value to a bounded range, and **SelfConsistency** measures the fraction of LLM generations that agreed on the same value among multiple independent sampling iterations.

To ensure interpretability and stable scaling, we normalize raw log-probabilities using a temperature-controlled sigmoid function. We set the temperature $\tau = 0.5$ to avoid saturation and preserve sensitivity in the moderate-confidence regime. We set $w_{\text{logp}} = 0.7$ and $w_{\text{cons}} = 0.3$ to prioritize the log-probability signal while still leveraging the stabilizing effect of self-consistency. These weights were empirically determined via a grid search on a validation set of 10 FM records with manually verified ground truth. We optimized for maximum agreement between the confidence score and human verification decisions, using the area under the precision-recall curve (AUC) as the selection criterion. We observed that prioritizing the log-probability signal improved precision, while incorporating self-consistency helped identify low-confidence outliers. However, these weights are not necessarily fixed and can be adjusted by users depending on the properties of their LLMs, model domains, or confidence calibration needs. Any field with a final confidence below a threshold $\theta = 0.75$ is automatically flagged for human review. Importantly, annotators inspect only the flagged fields rather than full model records. Reviewing all fields for all FMs would require substantial annotation effort, as each record contains many heterogeneous metadata elements. To assess the risk of confidently incorrect extractions, we manually inspected all fields for 10 records and found high-confidence outputs to be consistently accurate, supporting the reliability of our scoring mechanism. In practice, occasional field-level errors have limited impact on FM selection aas the most decisive properties (modality, architecture, compute requirements, and performance) are usually clearly stated and rarely mis-extracted.

**Diversity of Coverage.** The current release of **RS-FMD** spans a broad range of RSFMs pretrained on various data modalities (multispectral, hyperspectral, SAR, LiDAR, and text) and employing diverse model architectures (transformer-based encoders, CNN–transformer hybrids, vision–language models). Pretraining data sources range from small curated datasets to million-scale image collections, and spatial resolutions span from sub-meter imagery to coarse multispectral composites. By consolidating these heterogeneous information into a schema-guided resource, **RS-FMD** supports reproducible comparison, systematic benchmarking, and agent-compatible retrieval. We will maintain **RS-FMD** by hosting on a public repository under a permissive license. We periodically monitors new RSFM releases and inserts verified entries. To support broader scalability, we are developing a user interface where model authors can upload documentation of new models. The system will automatically extract metadata and present it to authors for correction before submission. We will review community-submitted updates to ensure consistency and reliability.

## 4 REMSA AGENT ARCHITECTURE

The goal of **REMSA** is to automate the selection of FMs for RS tasks through a reasoning-centered, modular agentic workflow. **REMSA** integrates structured knowledge grounding, LLM-based ranking with in-context learning, and iterative clarification to produce transparent and reproducible selections. Selecting an appropriate RSFM is challenging, as the models differ in data modalities, pretraining strategies, benchmark

performance, and resource requirements. In addition, users often provide incomplete or ambiguous task descriptions, requiring the agent to interpret intent and reconcile trade-offs among candidate models. To address these challenges, REMSA provides an integrated pipeline combining different agent components and external tools. This pipeline can achieve different targets such as structured retrieval, ranking, clarification, and memory archiving under a customized orchestration mechanism. This section will describe the agent workflow and the details of each component and tool.

## 4.1 AGENT WORKFLOW

Figure 1 illustrates the architecture of REMSA. The system is composed of two main layers: the **LLM agent core** and a set of **external tools**. The LLM agent core consists of two key components: the *Interpreter*, which parses user inputs into structured constraints and extracts user intent, and the *Task Orchestrator* dynamically decides which external tool to invoke at each step based on the current task state. When a user submits an free-text query, the query parser transforms it into a structured representation of constraints. We prompt the LLM with a carefully designed schema that covers both mandatory and optional fields relevant to RSFM selection (See Appendix B. for complete schema.). Specifically, the parser extracts the target application(e.g., land cover classification, surface water segmentation) and the required modality(e.g., multispectral, SAR) as mandatory fields to narrow the FM search space. Then REMSA integrates broader practical constraints through optional fields and clarification steps, including data availability, compute budget, fine-tuning requirements, and output quality priorities. Once constraints are available, the *Task Orchestrator* initiates a control loop that manages the entire selection process. At each step, it first evaluates the current task state, i.e., which constraints are available, how many candidates remain, and how confident the system is. Then it invokes the appropriate tool accordingly. If no mandatory constraints are missing, the orchestrator calls the *Retrieval Tool* to generate an initial candidate set. If the candidate set is small and all constraints are satisfied, the *Ranking Tool* is applied directly. If there are too many candidates or if ranking results yield low confidence scores, the orchestrator calls the *Clarification Generator Tool* to ask the user for additional input. The updated query is then passed back through the same loop. Once the top-$k$ result is obtained, the *Explanation Generator Tool* is invoked to produce the final report. This decision-making process is executed by empirical thresholds for ranking confidence, constraint coverage, and clarification rounds. The orchestration ensures that tool invocation is adaptive, goal-oriented, and transparent. To support personalization and self-improvement, REMSA also integrates *Task Memory*, which stores past user interactions in a vector database. Relevant memory entries are retrieved via cosine similarity to improve future interactions. More details on the implemented workflow are in Appendix C.. To enhance REMSA's reliability, we have several built-in mechanisms to mitigate failures. The orchestrator monitors confidence signals and triggers clarification rounds when ranking is uncertain. Rule-based constraint eliminates candidates that violate hard requirements. A fallback "closest-match" mode returns the safest alternative when no candidate fully satisfies the constraints. Our modular design also allows for integrating explicit feedback mechanisms (e.g., an LLM-as-a-Judge component that re-evaluates low-quality selections), making REMSA extensible to more robust self-correction strategies.

## 4.2 AGENT TOOLS

The following tools operate as callable interfaces outside of the agent core. Each tool is invoked independently by the orchestrator, depending on the state of the task, supporting retrieval, ranking, clarification, and explanation within the RSFM selection workflow. Our design supports extensibility for tool integration.

**Retrieval Tool.** To generate an initial set of candidates, the retrieval tool encodes both the structured user constraints and the FM entries in the **RS-FMD** using Sentence-BERT embeddings (Reimers & Gurevych, 2019). To preserve the structure of the metadata in the embedding, each metadata field is prefixed with a token of the type-indicator (e.g., [APPLICATION], [MODALITY]) before encoding. REMSA uses Facebook

AI Similarity Search (FAISS) (Meta, 2025) for an efficient approximate search based on cosine similarity. The tool returns a list of the most relevant FMs determined by a configurable similarity threshold. User can adjust it based on their domain requirements. In our experiments, we set this threshold empirically to ensure broad coverage while minimizing irrelevant matches. This tool is optimized for high recall: it includes soft matches and does not enforce strict constraints, allowing the downstream pipeline to handle finer filtering.

**Ranking Tool.** While the retrieval tool provides a broad list of relevant FMs, it cannot fully capture user-specific needs and deployment trade-offs. This task can be handled by a ranking tool. The ranking tool refines the candidate FM list using a hybrid strategy to balances efficiency, flexibility, and interpretability:

- *Rule-Based Filtering:* Candidates that violate hard constraints, such as required modality, sensor support, or minimum performance, are eliminated using deterministic logic. These hard constraints are defined based on fields extracted by the interpreter.
- *In-Context LLM Ranking:* The remaining candidates are re-ranked by an LLM prompted with the structured query and FM metadata, using expert-crafted few-shot examples to illustrate selection. The LLM returns an ordered list with brief justifications, leveraging in-context reasoning without any model training (details in Appendix D.). We also compute a confidence score for each selection following Section 3.

**Clarification Generator Tool.** If the orchestrator detects insufficient constraints or a low overall confidence score of selected FMs, it invokes the clarification tool. This tool inspects the parsed schema to determine missing or underspecified fields (e.g., modality, region, or performance bounds) and formulates clarification questions. The tool generates each question based on the interpreter schema. We limit the clarification to three rounds to avoid user fatigue. The agent will integrate the responses with initial user input, parse and merge them into the evolving task specification, in order to iteratively refine the selection process.

**Explanation Generator Tool.** Once a ranking is available, this tool generates structured, human-readable explanations. It uses a prompt-driven LLM to synthesize the rationale for each selected FM, including key reasons considering suitability and trade-offs. Each output includes the model name, bullet points for explanation, and links to the corresponding paper and code repository. This tool enhances transparency and user trust by exposing the decision process (prompt is in Appendix E.). The output is in JSON format.

## 5 Evaluation Protocol and Benchmark

Evaluating FM selection in RS is challenging due to the lack of dedicated benchmarks. Previous works mainly focus on evaluating model performance on fixed applications or datasets, rather than assessing the ability to recommend the most suitable FM under diverse real-world deployment constraints. In this work, we leverage **RS-FMD** to construct the first agent-oriented benchmark for FM selection, systematically covering diverse models, modalities, and deployment constraints.

**Benchmark Construction.** Our evaluation protocol relies on structured expert review, ensuring methodological rigor without imposing excessive annotation overhead. We curate a benchmark of 75 natural language queries to keep evaluation feasible while still ensuring diversity. We will publish these queries in our repository. All model-query pairs were evaluated by two experts from a computer science background with expertise in RS. We used a structured rubric to ensure consistency. Full details of the expert procedure are provided in Appendix G.. The evaluation resulted in 900 expert ratings as we compare the top-3 FMs from **Remsa** and 3 baselines. Each instance must be carefully rated across seven criteria. Thus, the evaluation workload is substantial despite the modest query count. To maximize representativeness, we create the query using structured templates of various scenarios and instantiate them (full templates is in Appendix H..) The queries diverse over data availability, computational resources, application complexity, and evaluation priorities. The resulting queries cover a wide range of tasks, including flood mapping with SAR data, crop type classification using multispectral or hyperspectral imagery, urban expansion monitoring with optical time

Table 1: Expert evaluation criteria.

| Criterion | Description |
|---|---|
| Application Compatibility | Whether the model fits the user requested application |
| Modality Match | Whether the model supports the required input data modality |
| Reported Performance | Performance reported on similar datasets or applications |
| Efficiency | Suitability for the user's computational resources |
| Popularity | Based on GitHub repository stars and citations |
| Generalizability | Diversity and scale of pretraining data |
| Recency | Whether the model reflects recent developments |

series, and disaster response, such as sea ice and wildfire detection. These tasks cover both single-date and multi-temporal analysis, single- and multi-modal inputs, and varying resource environments. All queries were reviewed by a domain expert for factual accuracy and corrected for consistency.

**Baselines.** There is no prior work on automated FM selection for RS deployment, and existing AutoML or agent systems cannot directly perform this task. We therefore design baselines that serve as both meaningful comparisons and implicit ablations of **REMSA**, with each baseline removing or modifying specific components to assess their contributions:

1. **REMSA-NAIVE**: Same toolset and DB as **REMSA**, but employs only basic sequential orchestration without out **REMSA**'s adaptive, task-aware control logic. It relies on LangChain's default single-step execution, where the LLM independently chooses tools without structured workflow or multi-turn coordination (LangChain, 2025). This baseline tests the effectiveness of our orchestration mechanism.
2. DB-RETRIEVAL: Returns the top-$k$ models from the FAISS-based dense retrieval over **RS-FMD**, with ranking, clarification, memory, and orchestration removed. This serves as a retrieval-only baseline and isolates the contribution of LLM-based ranking and constraint reasoning.
3. UNSTRUCTURED-RAG: A generic RAG setup where the LLM receives the query and raw, unstructured FM descriptions and outputs top-$k$ FMs with brief justifications (prompt in Appendix F.). This baseline tests whether LLM can perform FM selection without our structured, modular agent.

**Evaluation Protocol and Criteria.** For each query, **REMSA** and all baselines output their top-3 FM selections. These model-query pairs were then evaluated independently and blindly by the two experts using the criteria in Table 1. After individual scoring, disagreements were resolved through rubric-guided discussion. The evaluation was performed once during the scoring, and no adjustments were made to any FMs thereafter to avoid introducing bias. Each FM is rated on a 1-5 scale (0.5 precision) on 7 criteria in Table 1, covering task relevance and deployability under real-world constraints. Several criteria use explicit rules. For example, generalizability combines geographic, modality, and dataset-scale factors, popularity relies on citations or GitHub activity, and recency is based on publication year. They are designed to be transparent, reproducible, and grounded in practical needs rather than ad-hoc user preferences. More details on the evaluation procedure are in Appendix G.. The final score is a weighted sum of all criteria ratings (Our weight setting is in Appendix I.). The score is linearly mapped to 1-100 scale to better show the differences.

Although exhaustive empirical benchmarking of all candidate models is infeasible, our protocol offers a reproducible and practical proxy for evaluating agent performance in real-world FM selection workflows. To support transparency and broader community adoption, we publicly release the full set of evaluation queries, expert guidelines, scoring criteria, and model metadata used in the evaluation. This enables reproducibility and provides a standardized foundation for future research on FM selection in RS and beyond. Our evaluation does not assume a single ground-truth "best" FM. Experts compare the top-ranked candidates from all systems, and a system is preferred when its top model is judged more suitable than other systems. **REMSA** returns top-$k$ FMs with explanations, enabling users to choose based on their own preferences.

Table 2: Comparison to the baselines.

| System | Avg Top-1 | Avg Set | Top-1 Hit | HQ Hit | MRR |
|---|---|---|---|---|---|
| **REMSA** (Ours) | **75.76** | **75.03** | 22.67% | **40.00%** | **0.38** |
| **REMSA**-NAIVE | 72.67 | 72.00 | **25.33%** | 37.33% | 0.36 |
| DB-RETRIEVAL | 67.37 | 68.78 | 13.33% | 17.33% | 0.25 |
| UNSTR.-RAG | 71.23 | 68.39 | 13.33% | 30.67% | 0.24 |

Table 3: Sensitivity analysis on evaluation criteria.

| Criteria Setting | Avg Set | Top-1 Hit | MRR | Note |
|---|---|---|---|---|
| Full Scoring (All Criteria) | 75.03 | 22.67% | 0.38 | |
| w/o Application Compatibility | 73.32 | 21.33% | 0.36 | Green: |
| w/o Modality Match | 70.88 | 22.67% | 0.36 | Increase |
| w/o Reported Performance | 75.05 | 22.67% | 0.38 | Red: |
| w/o Efficiency | 80.23 | 25.33% | 0.38 | Drop |
| w/o Popularity+Recency | 75.13 | 25.33% | 0.39 | |
| w/o Generalizability | 75.10 | 22.67% | 0.38 | |

# 6 RESULTS AND ANALYSIS

We conduct experiments to comprehensively evaluate the effectiveness of **REMSA** in RSFM selection. Since no prior work directly targets real FM selection under diverse deployment constraints, we develop our own baselines. This section presents our experiment setup, quantitative results, and sensitivity analysis, followed by a discussion of limitations and representative examples.

**Experimental Setup.** We use `GPT-4.1` (OpenAI, 2025) for **REMSA** core and all baselines to ensure fairness. However, we design **REMSA** to be LLM-agnostic and support any LLM (e.g., `DeepSeek-R1` (DeepSeek-AI et al., 2025), `LLaMA-3` (Dubey et al., 2024)). Our benchmark consists of 75 diverse natural language user input queries. For each input, **REMSA** and all baselines (all described in Section 5) select the top-3 candidate FMs for comparison. Domain experts rate each candidate using the criteria in Table 1, and we report both single-model and set-level scores to evaluate selection accuracy and reasoning quality across multiple agent decision points. During evaluation, all clarification rounds in **REMSA** were executed automatically, with the system interacting with an independent LLM simulating user responses. No human was involved in these interactions, ensuring consistency and preventing evaluator bias.

**Evaluation Metrics.** We use complementary metrics to evaluate both the best model and the overall set quality: (1) *Average Top-1 Score* (expert score of the top-ranked model), (2) *Average Set Score* (average score of the top-3 models), (3) *Top-1 Hit Rate* (fraction where the system's top model is the expert's highest-scored), (4) *High-Quality Hit Rate* (fraction where the top model scores $\geq 80$), and (5) *Mean Reciprocal Rank - MMR* (rank of the expert-preferred model within the top-3).

## 6.1 COMPARISON TO BASELINES

As shown in Table 2, **REMSA** consistently outperforms all baselines in all evaluation metrics, demonstrating its effectiveness in selecting FMs under various real constraints. Illustrative examples of expert-scored model-query pairs are provided in Appendix J.. **REMSA** achieves the highest Average Top-1 Score (75.76) and Average Set Score (75.03), indicating not only that the top-ranked models are aligned with expert preferences, but also that the top-3 selections offer strong and diverse alternatives. Compared to DB-RETRIEVAL, which relies on similarity-based retrieval over structured metadata, **REMSA** improves Top-1 Hit Rate from 13.33% to 22.67%, and MRR from 0.25 to 0.38. This underscores the value of reasoning beyond retrieval, especially when user queries involve constraints (e.g., modality, resolution, compute budget) not explicitly stored in the metadata. Although UNSTRUCTURED-RAG has access to full model descriptions, its performance remains lower due to the lack of structured guidance and modular reasoning. This result shows that **REMSA**'s ability to combine structured schema grounding with dynamic tool orchestration enables precise alignment with user needs. Both **REMSA** and **REMSA**-NAIVE perform notably better than retrieval-only or unstructured RAG baselines, showing effectiveness of our agent architecture: grounding the selection process in a structured schema and enabling tool-based reasoning provides a substantial advantage. However, **REMSA** improves further in all major evaluation metrics. Although **REMSA** has a slightly lower Top-1 Hit Rate (22.67% vs. 25.33%), the higher Average Top-1 Score (75.76) and MRR (0.38) suggest that **REMSA** selects high-quality models more consistently at the top of its ranking. This indicates that our orchestration

logic, including multi-turn clarification and reasoning heuristics, contributes meaningfully to performance, especially when model choices are ambiguous or task formulations are complex.

**Latency Trade-off.** To assess the latency-performance trade-off, we measure the average end-to-end runtime per query. As expected, single-step methods are faster: DB-Retrieval takes 0.77s, Unstr.-RAG 11.9s, and **REMSA**-Naive 22.7s, whereas **REMSA** requires **31.7s** due to multi-stage reasoning and optional clarification. Despite this moderate overhead, **REMSA** delivers the highest expert-validated accuracy across major metrics, indicating that its additional reasoning steps yield meaningful and consistent gains.

## 6.2 SENSITIVITY ANALYSIS ON EVALUATION CRITERIA

To understand how well **REMSA** aligns with expert-defined evaluation principles, we perform a sensitivity analysis by removing each scoring criterion individually from the expert evaluation protocol. As shown in Table 3, the performance is generally robust in most dimensions, but some removals reveal important insights into which criteria contribute the most to the effective selection of the model. Both the removal of Application Compatibility and Modality Match lead to notable performance drops, confirming that **REMSA** actively prioritizes functionally appropriate models aligned with the user's objective. Notably, removing Reported Performance and Generalizability yields minimal change in overall results, implying that these dimensions are either captured implicitly through other criteria or are less decisive in the current benchmark setup. In contrast, removing Efficiency or Popularity+Recency actually leads to a modest performance gain. This suggests that while these criteria add practical relevance for deployment, they may occasionally favor well-known or resource-efficient models over technically optimal ones. The sensitivity results further validate that **REMSA** does not overfit to superficial indicators such as citations or recency, but instead emphasizes core compatibility and reasoning in its final decisions.

## 7 CONCLUSION AND DISCUSSION

We proposed **REMSA**, the first LLM Agent combine a FM database for real RSFM selection problems. By orchestrating modular tools for metadata retrieval, in-context ranking, multi-turn clarification, and memory-augmented reasoning, **REMSA** delivers high-quality and consistent selections. A key foundation is our **RS-FMD**- the first database for RSFM. It consolidates heterogeneous descriptions into a structured form for transparent retrieval and comparison. On an expert-driven benchmark, **REMSA** outperforms retrieval-only, unstructured RAG, and naive agent baselines.

In future work, we plan to expand the benchmark to rarer and more complex scenarios, explore lightweight supervised enhancements, and improve explanation specificity and trustworthiness. We also aim to reduce expert burden by introducing semi-automated scoring and community-assisted annotation, which will make **RS-FMD** and the benchmark easier to extend to new FMs. In addition, we plan to adopt a mixed expert- and benchmark-based evaluation mechanism to further strengthen robustness. We further envision extending **REMSA** toward adaptive decision-making, where the agent not only selects but also recommends model adaptation strategies, such as fine-tuning or domain-specific adjustment, and identifies opportunities for incorporating additional modalities when beneficial.

**Limitations.** Although **REMSA** performs successfully in the selection of RSFMs, some limitations remain: For example, our benchmark is based on 75 expert-annotated queries, which may miss rare or emerging use cases. However, the overall evaluation effort is substantial, totaling 900 expert ratings. In addition, the ranking relies on in-context learning rather than supervised training, which may limit performance on complex or uncommon queries. Despite these limitations, **REMSA** demonstrates the feasibility of constraint-aware agentic RSFM selection, setting the basis for future extensions to other scientific domains.

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

APPENDIX

## A. COMPLETE **RS-FMD** SCHEMA SPECIFICATION

To properly represent the properties of each FM, we designed a comprehensive data schema for **RS-FMD**. The schema includes the essential characteristics of model architectures, pretraining strategies, supported modalities, and benchmark performance.

Each model record includes fields such as unique identifiers, names, versions, release and update timestamps, and links to associated publications, code repositories, and pretrained weights. These metadata elements ensure traceability and reproducibility of the database entries.

Beyond these core descriptors, the schema incorporates detailed fields that capture architectural specifics (e.g., backbone type, number of layers, number of parameters), pretraining approaches (e.g., pretext training type, masking strategy), and modality integration. The design anticipates the diversity of RS models and supports future extensions.

To capture information about pretraining and evaluation comprehensively, the schema defines two nested structures:

- **PretrainingPhase**: This substructure records the datasets used for pretraining, geographical coverage, time range, image resolutions, token sizes, augmentation strategies, sampling methods, and masking ratios.
- **Benchmark**: This substructure captures evaluation metrics, including the applications, dataset, performance scores, and training hyperparameters used during evaluation.

Many fields are annotated with `free_text` metadata. This annotation signals that the field may contain natural language summarization that requires specialized treatment in confidence scoring and downstream verification.

Table 4 provides a comprehensive description of the fields of our data schema in **RS-FMD**, including nested structures for pretraining phases and benchmarks.

Table 4: Complete schema specification of **RS-FMD**, including nested pretraining phases and benchmarks.

| Field | Type | Description |
|---|---|---|
| *Main Model Fields* | | |
| model_id | string | Unique identifier of the model (free text) |
| model_name | string | Only the name of the model without extra descriptions (free text) |
| version | string | Version identifier (free text) |
| release_date | date | Release date of the model |
| last_updated | date | Last updated date |
| short_description | string | Short summary describing the model (free text) |
| paper_link | URL | URL to the associated publication |
| citations | integer | Number of citations |
| repository | URL | URL to the code repository |
| weights | URL | URL to pretrained model weights |
| backbone | string | Specific backbone used (free text) |
| num_layers | integer | Number of layers |
| num_parameters | float | Model size in millions of parameters |
| pretext_training_type | string | Type of pretext training strategy (free text) |

| Field | Type | Description |
|---|---|---|
| masking_strategy | string | Masking strategy applied during training (free text) |
| pretraining | string | Description of pretraining approach (free text) |
| domain_knowledge | list[string] | Domain-specific knowledge or methods incorporated |
| backbone_modifications | list[string] | Modifications made to the backbone |
| supported_sensors | list[string] | Supported satellite sensors |
| modality_integration_type | string | Integration type (free text) |
| modalities | list[string] | Input data modalities (free text) |
| spectral_alignment | {full, partial, none} | Whether the model models spectral continuity |
| temporal_alignment | {full, partial, none} | Whether the model models temporal sequences |
| spatial_resolution | string | Spatial resolution of data (free text) |
| temporal_resolution | string | Temporal resolution of data (free text) |
| bands | list[string] | Spectral bands used |
| *Nested: PretrainingPhase* | | |
| dataset | string | Dataset used for pretraining (free text) |
| regions_coverage | list[string] | Geographical regions covered |
| time_range | string | Time range of pretraining data (free text) |
| num_images | integer | Number of images used |
| token_size | string | Token size (free text) |
| image_resolution | string | Input image resolution (free text) |
| epochs | integer | Number of epochs |
| batch_size | integer | Batch size |
| learning_rate | string | Learning rate (free text) |
| augmentations | list[string] | Augmentations applied |
| processing | list[string] | Additional preprocessing steps |
| sampling | string | Sampling strategy (free text) |
| processing_level | string | Processing level (free text) |
| cloud_cover | string | Cloud cover filtering (free text) |
| missing_data | string | Handling of missing data (free text) |
| masking_ratio | float | Masking ratio |
| *Nested: Benchmark* | | |
| application_type | string | Type of application evaluated (free text) |
| application | string | Specific application domain (free text) |
| dataset | string | Benchmark dataset name (free text) |
| metrics | list[string] | List of evaluation metrics |
| metrics_value | list[float] | Numeric values for each metric |
| sensor | list[string] | Sensors used |
| regions | list[string] | Regions evaluated |
| original_samples | integer | Total number of samples before sampling |
| num_samples | integer | Actual number of samples used |
| sampling_percentage | float | Fraction of dataset retained (0–100) |
| num_classes | integer | Number of classes |
| classes | list[string] | Names of each class |
| image_resolution | string | Input image resolution (free text) |
| spatial_resolution | string | Spatial resolution (free text) |
| bands_used | list[string] | Bands used during evaluation |

| Field | Type | Description |
|---|---|---|
| augmentations | list[string] | Data augmentations applied |
| optimizer | string | Optimizer used (free text) |
| batch_size | integer | Batch size |
| learning_rate | float | Learning rate |
| epochs | integer | Number of epochs |
| loss_function | string | Loss function (free text) |
| split_ratio | string | Train/val/test split ratio (free text) |

Below we include a complete example of an **RS-FMD** record for the RSFM *A2-MAE*. This illustrates how the schema is instantiated with real metadata.

```
1  {
2    "model_id": "A2-MAE",
3    "model_name": "A2-MAE",
4    "version": "v1",
5    "release_date": "2024-06-16",
6    "last_updated": "2024-06-16",
7    "short_description": "A2-MAE is a spatial-temporal-spectral unified remote
        sensing pre-training method based on an anchor-aware masked autoencoder. It
          leverages a global-scale, multi-source dataset (STSSD) and introduces an
        anchor-aware masking strategy and a geographic encoding module to
        efficiently integrate spatial, temporal, and spectral information from
        diverse remote sensing imagery.",
8    "paper_link": "https://arxiv.org/abs/2406.08079",
9    "citations": 7,
10   "backbone": "ViT-Large",
11   "pretext_training_type": "Masked Autoencoder (MAE) with anchor-aware masking
        and geographic encoding",
12   "masking_strategy": "Anchor-aware masking (AAM): dynamically adapts masking
        ...",
13   "pretraining": "Self-supervised pre-training on the STSSD dataset...",
14   "domain_knowledge": [
15     "Geographic encoding (latitude, longitude, GSD)",
16     "Spatial-temporal-spectral relationships",
17     "Clustering-based data pruning"
18   ],
19   "supported_sensors": [
20     "Sentinel-2", "Landsat-8", "Gaofen-1", "Gaofen-2"
21   ],
22   "modality_integration_type": "Homogeneous Multimodal",
23   "modalities": ["Multispectral", "Multi-temporal"],
24   "spectral_alignment": "partial",
25   "temporal_alignment": "partial",
26   "spatial_resolution": "0.8-30m",
27   "temporal_resolution": "2020-2023, periodic seasonal revisits",
28   "bands": [
29     "Sentinel-2: B1-B12",
30     "Landsat-8: B1-B7",
31     "Gaofen-1: B1-B4",
32     "Gaofen-2: B1-B4"
33   ],
34   "pretraining_phases": [
```

```
35      {
36        "dataset": "STSSD",
37        "regions_coverage": ["Global (12k urban centers, 10k nature reserves)"],
38        "time_range": "2020-2023",
39        "num_images": 2500000,
40        "token_size": "16x16",
41        "image_resolution": "0.8-30m (cropped 256x256 to 3200x3200)",
42        "epochs": 130,
43        "batch_size": 1024,
44        "learning_rate": "1e-4 (cosine decay)",
45        "processing": [
46          "Atmospheric/radiation correction",
47          "Pan-sharpening (Gaofen)",
48          "Cropping/resizing alignment"
49        ],
50        "sampling": "Clustering-based pruning (keep hardest 10%)",
51        "cloud_cover": ">=10%",
52        "masking_ratio": 0.75
53      }
54    ],
55    "benchmarks": [
56      {
57        "task": "Classification",
58        "application": "Land cover classification",
59        "dataset": "EuroSAT",
60        "metrics": ["Accuracy"],
61        "metrics_value": [99.09],
62        "sensor": ["Sentinel-2"],
63        "regions": ["34 European countries"]
64      },
65      {
66        "task": "Classification",
67        "application": "Multi-label classification",
68        "dataset": "BigEarthNet",
69        "metrics": ["mAP"],
70        "metrics_value": [83.0]
71      },
72      {
73        "task": "Segmentation",
74        "application": "Surface water segmentation",
75        "dataset": "Sen1Floods11",
76        "metrics": ["mIoU"],
77        "metrics_value": [88.87]
78      },
79      {
80        "task": "Segmentation",
81        "application": "Cropland segmentation",
82        "dataset": "CropSeg",
83        "metrics": ["mIoU"],
84        "metrics_value": [44.81]
85      },
86      {
87        "task": "Change Detection",
88        "application": "LEVIR-CD",
89        "dataset": "LEVIR-CD",
```

```
 90        "metrics": ["mIoU"],
 91        "metrics_value": [84.32]
 92      },
 93      {
 94        "task": "Change Detection",
 95        "application": "Urban change detection",
 96        "dataset": "OSCD",
 97        "metrics": ["F1"],
 98        "metrics_value": [53.97]
 99      },
100      {
101        "task": "Change Detection",
102        "application": "Semantic change segmentation",
103        "dataset": "DynamicEarthNet",
104        "metrics": ["mIoU"],
105        "metrics_value": [46.0]
106      }
107    ]
108  }
```

## B.  STRUCTURED QUERY SCHEMA

Below we show the complete JSON schema template used by the query interpreter:

```
{
  "application": "string",                         // Mandatory
  "modality": "string",                       // Mandatory
  "sensor": "string or list of strings",      // Optional
  "spatial_resolution": "string or numeric",  // Optional
  "temporal_resolution": "string or numeric", // Optional
  "bands": "list of strings",                 // Optional
  "avaliable_data": "string",                 // Optional
  "deployment_device": "string",              // Optional
  "priority_metrics": "list of string",       // Optional
  "min_performance": {                        // Optional
    "metric": "list of string",
    "value": "list of number"
  },
  "region": "string or list of strings",      // Optional
  "domain_keywords": "list of strings"        // Optional
}
```

## C.  IMPLEMENTATION DETAILS

---

**Algorithm 1: REMSA Workflow for RSFM Selection**

---

**Input:** User Query $q$, desired number of recommendations $k$
**Output:** Top-$k$ selected models with explanations

1 Initialize $ClarifyCounter \leftarrow 0$
2 Initialize $MaxClarify \leftarrow 3$
3 **repeat**
4     $Constraints \leftarrow$ **ParseQuery**($q$) ;              // LLM parses constraints
5     **if** *mandatory constraints missing* **then**
6         **if** $ClarifyCounter < MaxClarify$ **then**
7             $q \leftarrow$ **ClarifyUser**($q, Constraints$) Increment $ClarifyCounter$
8         **else**
9             **break** ;             // Stop clarifying to avoid user fatigue
10 **until** All mandatory constraints are present;
11 $Candidates \leftarrow$ **RetrieveModels**($q$) ;           // Embedding retrieval (Top K)
12 $Filtered \leftarrow$ **FilterCandidates**($Candidates, Constraints$)
13 **if** $|Filtered| = 0$ **then**
14     $BestMatch \leftarrow$ **SelectClosestModel**($Candidates, Constraints$)
15     $Explanation \leftarrow$ **GenerateExplanation**($q, BestMatch$)
16     **return** {Recommendation: BestMatch, Explanation}
17 **if** $|Filtered| > MaxCandidates$ **then**
18     **if** $ClarifyCounter < MaxClarify$ **then**
19         $q \leftarrow$ **ClarifyUser**(q, Constraints)
20         Increment $ClarifyCounter$
21         **Go to line 3** ;          // Restart process with clarified query
22 $Scores \leftarrow$ **RankCandidates**($q, Filtered$) $OverallConfidence \leftarrow$ **ComputeConfidence**($Scores$)
23 **if** $OverallConfidence < ConfidenceThreshold$ **then**
24     **if** $ClarifyCounter < MaxClarify$ **then**
25         $q \leftarrow$ **ClarifyUser**($q, Constraints$)
26         Increment $ClarifyCounter$
27         **Go to line 3**
28 $TopK \leftarrow$ Top-$k$ candidates in $Filtered$ ranked by $Scores$
29 $Explanation \leftarrow$ **GenerateExplanation**($q, TopK$)
30 **return** {Recommendations: TopK, Explanation}

---

The workflow of **REMSA** is shown in Algorithm 1. The pipeline is implemented in Python using `pydantic` for schema validation, and the OpenAI GPT-based models for extraction. Each input document is processed in multiple iterations to collect diverse generations. The **RS-FMD** is stored in JSONL records and versioned via DVC to ensure reproducibility.

## D.  LLM-BASED IN-CONTEXT RANKING PROMPT

To re-rank candidate foundation models without training a dedicated learning-to-rank model, we leverage in-context learning (ICL) with a LLM. The prompt explicitly instructs the LLM to prioritize user requirements, compare candidate models, and produce a ranked list with explanations. We provide few-shot examples

created by an expert in the prompt to guide the model toward consistent ranking behavior. The prompt is connected to **RS-FMD** to provide the metadata of the candidate models. Below is the prompt template we are using in the ranking module:

**Prompt Template:**

```
You are an expert in remote sensing foundation model selection.

You will be given:
1. A structured user query specifying task requirements and constraints.
2. A list of candidate models retrieved from a database, each with metadata
    fields.

Your goal:
- Rank the candidate models from most to least suitable for the user's query.
- For each model, provide a brief explanation in several bullet points
    describing why it is placed at that rank.
- Prioritize hard constraints (application, modality, required sensor, and
    min_performance if provided), then consider secondary preferences (spatial/
    temporal resolution, application type, domain keywords, etc.).
- When two models equally satisfy the constraints and preferences, prefer the
    model that is more efficient, better validated on diverse benchmarks, or
    more versatile(multimodal, multi-temporal).

[Example]
Structured Query:
{
  "application": "land cover classification",
  "modality": "multispectral",
  "sensor": ["Sentinel-2"],
  "min_performance": {
    "metric": ["accuracy"],
    "value": [85]
  }
}

Candidate Models:
1. S2MAE
2. Prithvi
3. CACo

Ranking Output:
1. S2MAE
   - Directly supports Sentinel-2 multispectral data
   - Achieves 99.1\% accuracy on EuroSAT, exceeding 85\% requirement
   - Purpose-built for land cover classification
2. Prithvi
   - Supports multi-temporal multispectral data, including Sentinel-2
   - Accuracy slightly below requirement on similar tasks
   - More generalist FM
3. CACo
   - Only supports RGB modality
   - Accuracy below the 85\% requirement
   - Designed mainly for change detection and event retrieval

Your Task:
```

```
Given the following new query and candidates, produce a ranked list with
    explanations.

Structured Query:
{query}

Candidate Models:
{candidates}

Please output the ranked list as JSON in the following format:
[
  {
    "model": <model_name>,
    "rank": <integer>,
    "reason": [<short bullet points>]
  },
  ...
]
```

## E.   EXPLANATION GENERATOR PROMPT

The explanation generator uses an LLM to produce concise, interpretable justifications for the final ranked
FM list. The prompt template in our explanation generator is given as follows:

```
You are an expert in remote sensing foundation model selection.

The structured user query is:
{query}

The final ranked candidate models with their metadata are:
{ranked_models}

Your task:
1. For each model, output a JSON object with:
    - "model_name"
    - "explanation" (several bullet points on why it is recommended)
    - "paper_link"
    - "repository"
2. Highlight how the model satisfies or partially satisfies the query.
3. Mention key trade-offs if relevant (accuracy vs. efficiency, modality
coverage, etc.).
```

## F.   PROMPT FOR RAG-LLM BASELINE

For the LLM-RAG baseline, we prompt an LLM with the original user input and the retrieved model
documentation as a context. The LLM is instructed to select and rank the top three remote sensing foundation
models and provide concise explanations for each recommendation.

```
You are an expert in remote sensing foundation models.

The user has provided the following task description:
```

```
{user_input}

Below is a set of candidate models with their documentation:
{context_str}

Your task:
1. Select and rank the top 3 remote sensing foundation models most suitable for
    the task.
2. For each selected model, provide:
-- A short explanation of why it fits the task requirements.
-- The reason for its ranking position compared to others.
-- Any other relevant information from the context.
3. Follow this exact output format:

    1. model: <model_name>
    explanation:
    - <reason 1>
    - <reason 2>
    - <reason 3>

    2. model: <model_name>
    explanation:
    - <reason 1>
    - <reason 2>
    - <reason 3>

    3. model: <model_name>
    explanation:
    - <reason 1>
    - <reason 2>
    - <reason 3>
```

## G.  EXPERT EVALUATION PROCEDURE

**Expert Background.** All annotations were performed by two experts with a computer science background and specialization in RS. Both have prior experience working with RSFMs, have published in the relevant domains, and are familiar with model architectures, pretraining datasets, and evaluation practices.

**Annotation Protocol.** To ensure consistency and reproducibility, we followed a structured, multi-stage scoring protocol:

- **Rubric Design.** We created a detailed rubric for all seven criteria in Table 1, including definitions, examples, and decision rules.
- **Calibration Phase.** Both experts annotated an initial subset of model-query pairs. Disagreements were used to refine the rubric until interpretations aligned.
- **Independent and Blind Scoring.** Experts then rated all remaining model-query pairs independently and without access to system identities or each other's scores.
- **Disagreement Resolution.** Any pair with substantial disagreement was re-examined in a controlled discussion, with decisions resolved strictly according to the rubric.

**Objective Scoring Rules.** Where possible, we used explicit rules to reduce subjectivity:

- Reported Performance. Reported performance was determined by checking for benchmarks that matched the queried task. If none existed, we evaluated performance on broader but related tasks. For example, if the query specifies the task as scene classification, and there is no benchmark for this, we look for general classification benchmarks. Depending on its performance, this model gets a moderate/high reported performance score. Models with no relevant benchmarks received a low score.
- Efficiency. Model parameter counts were normalized to a 0-5 scale as a proxy for complexity, and combined with reported performance to obtain a final efficiency score. Specifically, we divide this complexity measure by the reported performance to produce a final efficiency score, also on a 0-5 scale. Popularity. Popularity was used as a practical usability indicator rather than a measure of inherent model quality. We used normalized GitHub star counts (when code exists) and Google Scholar citation counts (when paper is unavailable). This reflects maturity, community adoption, and available ecosystem support.
- Generalizability. We quantified pretraining diversity using three measurable components extracted from official FM documentation:
  1. Geographic diversity: global (score 5), multi-regional (3–4), or single-region coverage (1–2).
  2. Sensor-modality diversity: number of distinct modalities used in pretraining e.g., optical, SAR, multi-spectral, hyperspectral).
  3. Dataset scale: reported total area, number of scenes, or total images.

  These components were combined into a composite 1-5 score. Inter-annotator agreement confirmed that the rule-based definitions reduced subjectivity.
- Recency. Recency was defined by the publication year or the latest model-card update:

$$2025\text{–}2026 = 5, \quad 2024 = 4, \quad 2023 = 3, \quad 2022 = 2, \dots$$

  Given the rapid evolution in RSFMs, this criterion serves as a soft heuristic rather than a primary determinant.

**Reference Sources.** All judgments were grounded in publicly available references for each foundation model. Experts used: (1) published papers and preprints; (2) official GitHub repositories and model documentation; (3) public benchmark results; (4) citation databases; and (5) described pretraining datasets from official sources. These references provided the necessary information on modality support, reported performance, efficiency, generalizability, popularity, and recency.

## H. QUERY TEMPLATE FOR CREATING BENCHMARK DATASET

To construct a representative and diverse benchmark dataset for evaluation, we define 16 structured query templates. Each template corresponds to a specific category of user constraints:

- **Data Availability (A1–A5):**
  - A1: *No Training Data* — User wants to use pre-trained models directly.
  - A2: *Sufficient Labeled Data* — User has enough labels to fine-tune or train from scratch.
  - A3: *Few-shot Labels* — User has a small set of labeled data only and requires models that generalize in low-data regimes.
  - A4: *Unlabeled Data Only* — User has input data but no labels and seeks models suited for unsupervised or self-supervised settings.
  - A5: *Data Adaptation Needed* — User's data differs from typical inputs, requiring domain adaptation or compatibility adjustments.
- **Computational Resources (B1–B3):**
  - B1: *Limited Resources* — e.g., CPU-only laptop.
  - B2: *Moderate Resources* — e.g., desktop with GPU.
  - B3: *High Resources* — e.g., cluster-scale GPU compute.
- **Application Complexity (C1–C3):**

Table 5: Structured query templates used for benchmark dataset generation. Each template maps to one constraint category. Slot values ({application}, {sensor}, {region}) are drawn from a predefined vocabulary and paraphrased by an LLM.

| Template (Natural Language) | Categories |
|---|---|
| I'm looking for a model I can use out-of-the-box for {application} using {modality} data. I don't have any labeled training data. | A1 |
| I have a well-labeled dataset for {application} with {modality} in {region}. Which model would be best to fully fine-tune from scratch? | A2 |
| I only have a few labeled samples for {application} using {sensor}. I want a model that can adapt well in a few-shot setting. | A3 |
| I have a lot of unlabeled {modality} imagery from {region}. I need a model that works well with self-supervised or unsupervised learning for {application}. | A4 |
| My data uses {sensor} with {spatial_resolution} resolution, but most models I've seen don't support it. Can you recommend one that can be adapted? | A5 |
| I'm working on {application} but only have access to a laptop with no GPU. Which model would be small enough to run locally? | B1 |
| I'm using a desktop with a single GPU and doing {application} on {modality} imagery. Which models balance performance and efficiency? | B2 |
| For {application}, I have access to cloud GPUs and can afford large models. What's the most powerful foundation model I can try? | B3 |
| I'm doing basic {application} (e.g., 3–4 land classes). What lightweight model would you suggest for fast experimentation? | C1 |
| I'm working on multi-class classification {application} with {modality} images. The task isn't trivial, but I don't need pixel-level precision. | C2 |
| I need a model for high-resolution segmentation or fine-grained {application}. Accuracy and spatial detail are important. | C3 |
| For {application} using {sensor} data, I mainly care about achieving the highest overall accuracy, even if the model is large. | D1 |
| For {application} using {sensor} imagery, I want clean and accurate outputs with minimal false detections; clear boundaries and reliable predictions are most important. | D2 |
| For {application} using {sensor} imagery, I need to ensure all target instances are captured, even if some false alarms occur; completeness is critical. | D3 |
| I need fast inference for {application} in near real-time on {device}. What's a good lightweight model? | D4 |
| I'm doing {application} on {modality} in {region}, but I only have few-shot labels and limited compute. Which model fits this setup best? | Composite |

- C1: *Simple Application* — Applications with low label granularity or few classes (e.g., binary classification, basic change detection).
- C2: *Moderate Application* — Applications with moderate difficulty, such as multi-class classification or coarse semantic segmentation.
- C3: *Complex Application* — Applications requiring fine-grained spatial precision, multi-class segmentation, multi-modal fusion, or high-resolution outputs.
- **Evaluation Priorities (D1–D4):**
  - D1: *Accuracy-Focused* — Maximize correctness of classification or segmentation outcomes.
  - D2: *Output Quality-Critical* — Prioritize clean, well-bounded, and visually reliable outputs (e.g., high mIoU, sharp edges, no artifacts).
  - D3: *Coverage-Critical* — Ensure all relevant regions or objects are detected, even at the cost of some false positives (e.g., disaster mapping, change detection).

- D4: *Speed-Critical* — Require lightweight or low-latency models for fast inference on edge devices.

Accordingly, Table 5 shows the full list of templates used to generate the benchmark queries. Slot values (e.g., {application}, {sensor}, {region}) are drawn from a predefined vocabulary and instantiated using sampling and LLM-based paraphrasing.

## I.   EXPERT SCORING WEIGHT CONFIGURATION

To aggregate model evaluation scores during expert labeling, we apply a weighted linear combination of the seven criteria from Table 1. The weights are as follows:

| Criterion | Weight (%) |
|---|---|
| Application Compatibility | 25 |
| Modality Match | 20 |
| Reported Performance | 20 |
| Efficiency | 15 |
| Generalizability | 10 |
| Popularity | 5 |
| Recency | 5 |

These weights were empirically determined on the basis of expert interviews. We normalize raw scores before aggregation.

## J.   ILLUSTRATIVE EXAMPLES OF EXPERT SCORING

To improve transparency, we provide several examples demonstrating how experts applied the scoring rubric to real model-query pairs. Each example includes: (1) the natural-language query, (2) the top-3 FM selections from all systems, and (3) the expert ratings across the seven criteria defined in Table 1. These examples show how rubric-guided, independent scoring yields consistent and interpretable evaluations.

*Example 1:*

**Query:** *I need a model for fine-grained land cover classification using high-resolution multispectral imagery. Accuracy and spatial detail are important.*

**Selected FMs (Top-3 from Each System):** See Table 6.

*Example 2:*

**Query:** *I only have a few labeled samples for urban expansion detection using Sentinel-1 and Sentinel-2 time series data from 2016-2023. I want a model that can adapt well in a few-shot setting.*

**Selected FMs (Top-3 from Each System):** See Table 6.

These examples illustrate how the rubric was applied in practice and how expert judgments reflect both task requirements and model capabilities. They also demonstrate how rubric-guided scoring minimizes subjective variation across annotators.

Table 6: Evaluation results for queries 1 and 2. **Criteria:** CR1 - Application Compatibility; CR2 - Modality Match; CR3 - Reported Performance; CR4 - Efficiency; CR5 - Generalizability; CR6 - Popularity; CR7 - Recency.

| System | Rank | FM | CR1 | CR2 | CR3 | CR4 | CR5 | CR6 | CR7 | Final Score |
|---|---|---|---|---|---|---|---|---|---|---|
| | | | | | **Query 1** | | | | | |
| **REMSA** | 1 | OmniSat | 5 | 5 | 5 | 5 | 4 | 3 | 4 | 94 |
| | 2 | FlexiMo | 4 | 4.5 | 4 | 2.5 | 1.5 | 3.5 | 5 | 75 |
| | 3 | CtxMIM | 5 | 5 | 4.5 | 3 | 1.5 | 3.5 | 3 | 83.5 |
| **REMSA**-Naive | 1 | OmniSat | 5 | 5 | 5 | 5 | 4 | 3 | 4 | 94 |
| | 2 | FlexiMo | 4 | 4.5 | 4 | 2.5 | 1.5 | 3.5 | 5 | 75 |
| | 3 | CtxMIM | 5 | 5 | 4.5 | 3 | 1.5 | 3.5 | 3 | 83.5 |
| DB-Retrieval | 1 | SpectralEarth | 3 | 3 | 3.5 | 1.5 | 3 | 3 | 5 | 59.5 |
| | 2 | OmniSat | 5 | 5 | 5 | 5 | 4 | 3 | 4 | 94 |
| | 3 | MATTER | 4 | 4.5 | 4 | 4.5 | 3.5 | 1 | 2 | 75 |
| Unstr.-RAG | 1 | FoMo | 5 | 5 | 3.5 | 1.5 | 2 | 1.5 | 5 | 79.5 |
| | 2 | DynamicVis | 4 | 4 | 4 | 3.5 | 3.5 | 2 | 5 | 75 |
| | 3 | SatVision-TOA | 2.5 | 4 | 2.5 | 0 | 2.5 | 5 | 4 | 55 |
| | | | | | **Query 2** | | | | | |
| **REMSA** | 1 | SSL4EO-S12 | 5 | 5 | 4 | 4 | 4.5 | 4.5 | 3 | 89.5 |
| | 2 | Ial-SimCLR | 3.5 | 5 | 3.5 | 5 | 2 | 3 | 3 | 77.5 |
| | 3 | SeCo | 3 | 3 | 3.5 | 5 | 5 | 2.5 | 1 | 67 |
| **REMSA**-Naive | 1 | SoftCon | 5 | 5 | 4.5 | 3 | 3 | 4 | 4 | 87 |
| | 2 | SkySense | 5 | 5 | 5 | 1 | 3.5 | 5 | 4 | 85.5 |
| | 3 | SSL4EO-S12 | 5 | 5 | 4 | 4 | 4.5 | 4.5 | 3 | 89.5 |
| DB-Retrieval | 1 | CACo | 3 | 3 | 4 | 4 | 4 | 4 | 3 | 70 |
| | 2 | SeCo | 3 | 3.5 | 5 | 5 | 5 | 2.5 | 1 | 67 |
| | 3 | SSL4EO-S12 | 5 | 5 | 4 | 4 | 4.5 | 4.5 | 3 | 89.5 |
| Unstr.-RAG | 1 | CACo | 3 | 3 | 4 | 4 | 4 | 4 | 3 | 70 |
| | 2 | Copernicus-FM | 3 | 3.5 | 3 | 1 | 3.5 | 5 | 5 | 62.5 |
| | 3 | AnySat | 3.5 | 5 | 3.5 | 1.5 | 4 | 4.5 | 5 | 74 |