# OpenReview forum: "REMSA: An LLM Agent for Foundation Model Selection in Remote Sensing"
_ICLR.cc/2026/Conference — Submitted to ICLR 2026_

### Official Review · Reviewer_2USW · 2025-11-01

**Soundness:** 2
**Presentation:** 2
**Contribution:** 3
**Rating:** 4
**Confidence:** 4

**Summary:**

The authors propose REMSA, the first LLM agent that automates remote-sensing foundation-model selection from free-form queries. Powered by the schema-guided Foundation Model Database (FMD) catalog of 150+ models, it retrieves candidates, clarifies missing constraints, ranks choices via in-context learning and delivers transparent justifications. Evaluated on 75 expert-curated scenarios totaling 900 task-system-model ratings, REMSA surpasses dense-retrieval and RAG baselines, offering reproducible, user-tailored recommendations without accessing private data.

**Strengths:**

The work introduces the first domain-specific agent for remote sensing model selection, allowing natural language queries to yield ranked recommendations with explanations.
It builds FMD, a large, fully structured, and machine-readable database of remote sensing models, openly maintained for the community.
The authors propose a comprehensive evaluation protocol based on multiple real-world scenarios and expert scoring, showing that their agent-based approach outperforms retrieval-only and RAG baselines.

**Weaknesses:**

Evaluation is limited, using only a few experiments, without broad, multidimensional validation; code is currently unavailable.

**Questions:**

1.The framework relies on a closed-source commercial LLM (e.g., GPT-4.1), which raises concerns about reproducibility and accessibility. How can the authors demonstrate that REMSA remains effective when using open-source LLMs? Adding experiments with other LLMs would strengthen the evidence.

2.The baselines are entirely self-designed and do not include comparisons with established AutoML frameworks or human expert manual selection. How do the authors justify that these baselines are sufficiently strong, and what prevents the reported performance gains from being overstated?

---

> ### Author Response · Authors · 2025-11-20
> **Clarifying Details and Responding to All Questions (Updates to the paper are marked in blue text.)**
>
> Thank you for the constructive feedback. We improved the writing and address each comment below:
>
> **1. Evaluation is limited**
>
> **Answer:** We acknowledge that the current evaluation focuses on the core comparison but does not yet cover the full range of potential analyses. The current evaluation already demonstrates REMSA’s effectiveness through blinded expert comparison across four systems and seven evaluation criteria. These experiments validate the core claims of the paper and are sufficient to support the main conclusions. In recent RSFM studies such as GEO-Bench-2 [arXiv:2511.15658] also emphasize that comprehensive RSFM evaluation is resource-intensive and requires balancing breadth with annotation feasibility, which aligns with our design choices.
> However, to further strengthen the contribution, we are extending the validation with (1) a more fine-grained ablation analysis, (2) a latency-performance trade-off evaluation, and (3) cross-LLM tests using LLaMA-3 and DeepSeek-R1. **The results of (2) were added in Section 6.1. Others will be included in the final revision or the latest CR version.**
>
> **2. code+data is unavailable.**
>
> **Answer:** All data, FM metadata (FMD), benchmark queries, and code used in the paper were included at the submission time in the **supplementary material as a ZIP file**.
>
> **3. Compare different LLMs**
>
> **Answer:** While REMSA is designed to be LLM-agnostic, our main evaluation used GPT-4.1 because it was the most stable model available at the time of submission. To verify generalizability, we have also run preliminary tests on both LLaMA-3 and DeepSeek-R1 using a small subset of benchmark queries to validate the stability of REMSA's performance. However, we agree that a more systematic evaluation is needed. We are currently running the full set of experiments with multiple LLMs, and due to the time required for expert evaluation, **we will include the complete cross-LLM results in the final revision or latest CR version.**
>
> **4. baselines are self-designed. No compare to AutoML**
>
> **Answer:** Because no existing AutoML framework is designed for RSFM selection, there is unfortunately no established method that can serve as a direct baseline for our task. General AutoML frameworks (e.g., AutoKeras, AutoSklearn, H2O AutoML) focus on model training and hyperparameter optimization on a given dataset, and do not address FM selection, modality constraints, pretraining compatibility, or RS-specific considerations. As such, they cannot be applied meaningfully to the FM selection without building an entirely new interface and re-annotating training datasets, which is beyond the scope of our current work.
> Regarding human expert selection, we specifically do not use human selection as a baseline because human experts are already used as evaluators. Using the same experts as both judge and baseline would introduce bias.
> Instead, our baselines are designed to isolate the contribution of each component in REMSA: pure retrieval, unstructured RAG, and naive single-step agent orchestration. These represent the most feasible alternatives when no prior systems exist.
> **We added a short discussion for baseline selection in Section 5.**

---

### Official Review · Reviewer_ky4D · 2025-11-03

**Soundness:** 2
**Presentation:** 2
**Contribution:** 2
**Rating:** 4
**Confidence:** 4

**Summary:**

This paper introduces REMSA (Remote-sensing Model Selection Agent), a large language model (LLM)-based agent designed for automated selection of remote sensing foundation models (RSFMs). To address issues of scattered documentation, heterogeneous formats, and complex model selection in remote sensing, the authors constructed the RS Foundation Model Database (FMD)—the first structured database covering over 150 RSFMs, systematically organizing information on architectures, data modalities, training strategies, and benchmark performance. Building upon FMD, REMSA integrates structured retrieval, contextual ranking, multi-round clarification, and memory-augmented reasoning to achieve automated model recommendation from natural language task descriptions. The authors further established a benchmark with 75 expert-validated tasks and 900 configurations, and compared REMSA against baselines such as DB-Retrieval, Unstructured RAG, and Naive Agent. Experimental results demonstrate that REMSA significantly outperforms all baselines in Top-1 accuracy, MRR, and overall evaluation scores.

**Strengths:**

The main advantage of this paper lies in proposing the model selection problem in the remote sensing field, constructing a benchmark, and introducing the REMSA method.  REMSA is the first domain-specific LLM agent for RSFM selection, combining structured knowledge with reasoning capabilities to enable transparent and reproducible model recommendations. The FMD serves as the first structured RSFM resource, featuring an extensible schema and ensuring data quality through automated extraction and human verification. REMSA’s multi-round clarification and task-aware ranking mechanisms substantially enhance recommendation accuracy and interpretability.

**Weaknesses:**

Although the paper presents an innovative framework and system design, its experimental and evaluation components exhibit several limitations.
1. The experiments are clearly insufficient. The paper includes only a main experiment and a single ablation study, without conducting more comprehensive analyses or comparisons with existing retrieval-augmented or tool-scheduling approaches (e.g., ToolRerank [1], COLT [2]). As a result, the empirical evidence supporting REMSA’s relative performance and methodological advantages remains limited.
2. The proposed benchmark consists of only 75 query samples. While the authors explain that this decision was made to maintain the feasibility of expert evaluations, this scale is too small to capture the diversity and complexity of remote sensing tasks in foundation model selection. Moreover, both the benchmark and evaluation protocols suffer from limited extensibility. Although FMD can be updated with new foundation models, the current benchmark cannot directly incorporate them for unified evaluation. The evaluation process also depends heavily on expert judgments and does not cover all models in FMD, making it nearly impossible for other researchers to reuse the benchmark.
3. In addition, the paper does not provide any examples of expert scoring, which makes it difficult to assess the consistency and validity of the evaluation criteria. Including illustrative examples—such as a query, recommended models, and corresponding expert ratings—would greatly improve transparency and credibility. Finally, although the paper claims that REMSA is LLM-agnostic, the experiments are conducted solely with GPT-4.1, without testing other major models such as LLaMA-3 or DeepSeek-R1. This weakens the argument for the generalizability of the proposed framework.

[1] Toolrerank: Adaptive and hierarchy-aware reranking for tool retrieval. In Proceedings of the 2024 Joint International Conference on Computational Linguistics, Language Resources and Evaluation (LREC-COLING), 2024 124.
[2] Towards completeness-oriented tool retrieval for large language models. In: Proceedings of the 33rd ACM International Conference on Information and Knowledge Management. 2024, 1930–1940

**Questions:**

1. The experimental section appears somewhat limited, containing only a main experiment and a single ablation study, without comparisons to other retrieval-augmented tool selection approaches such as ToolRerank [1] or COLT [2]. Could the authors include additional comparative experiments to better demonstrate the empirical advantages of REMSA over existing methods?
2. Given that task templates can be readily expanded, it is unclear why the proposed benchmark includes only 75 queries. This number seems insufficient to capture the diversity and complexity of real-world remote sensing tasks. Could the authors explain the reasoning behind this limited scale?
3. The benchmark and evaluation framework appear to lack extensibility. New foundation models cannot be seamlessly integrated, and the expert-generated evaluation scores do not cover all models in the FMD, which limits reproducibility and external usability. How should other methods be evaluated when their results correspond to models not present in the current scoring set?
4. No examples of expert scoring are provided, making it difficult to assess the validity and consistency of the evaluation criteria. Could the authors provide representative examples of expert annotations—such as task queries, recommended models, and corresponding scores—in the appendix to enhance transparency?
5. Although the paper claims that the framework is LLM-agnostic, all experiments are conducted using GPT-4.1 without testing alternative models such as LLaMA-3 or DeepSeek-R1. Could the authors include additional experiments to verify the generalizability of the framework across different LLMs?

[1] Toolrerank: Adaptive and hierarchy-aware reranking for tool retrieval. In Proceedings of the 2024 Joint International Conference on Computational Linguistics, Language Resources and Evaluation (LREC-COLING), 2024 124.
[2] Towards completeness-oriented tool retrieval for large language models. In: Proceedings of the 33rd ACM International Conference on Information and Knowledge Management. 2024, 1930–1940

---

> ### Author Response · Authors · 2025-11-20
> **Clarifying Details and Responding to All Questions (Updates to the paper are marked in blue text.)**
>
> Thank you for the constructive feedback. We improved the writing and address each comment below:
>
> **1. compare with retrieval-augmented or tool-scheduling approaches**
>
> **Answer:** We examined whether tool-retrieval or tool-scheduling approaches such as ToolRerank or COLT could serve as baselines. However, these systems assume a library of independent, executable tools (e.g., APIs for calculation, search, translation) that an LLM can call in arbitrary order. In contrast, REMSA’s tools are not standalone APIs, but tightly coupled reasoning modules: schema parsing, constraint checking, retrieval, clarification, and ranking. These modules must be invoked in a fixed, interdependent workflow. They cannot be freely substituted, reordered, or selected via a tool-retrieval mechanism, and no dataset exists that maps FM-selection queries to per-module tool labels, which is required by such methods.
> Recent RSFM studies such as GEO-Bench-2 [arXiv:2511.15658] also highlight that RSFM evaluation requires structured, multi-stage reasoning across capability dimensions (modality, resolution, temporality, task type), rather than single-step tool invocation. This further supports the need for REMSA’s explicit orchestration rather than generic tool-scheduling methods. Applying ToolRerank or COLT would therefore be inappropriate and would require re-architecting both their methods and our pipeline.
>
> **2. Why only 75 queries?**
>
> **Answer:** Although the benchmark contains 75 queries, the total evaluation workload is much larger because every query must be evaluated across four systems, each returning three candidate models. This results in 900 expert evaluations. Each evaluation requires domain experts to carefully assess a model along 7 criteria based on the query, often requiring close reading of the model papers, checking benchmark results, and in some cases running small tests. Because each expert can only complete 2–3 evaluations per day at this level of depth, expanding the benchmark significantly would lead to an infeasible annotation effort. Our goal in this first version is to balance coverage and expert reliability: 75 queries allow us to span all template categories and major RS applications while keeping evaluation quality consistently high. In future work, we will extend the benchmark using a combination of additional expert annotations, semi-automated scoring pipelines, and community contributions. **This is discussed in Section 5.**
>
> **3. Limited extensibility of benchmark and evaluation protocols**
>
> **Answer:** Our benchmark is query-driven, which makes it possible to incorporate new FMs by simply re-running the systems and evaluating only the newly recommended model-query pairs. Because the scoring criteria are fixed and all queries are independent, existing expert scores remain valid and do not need to be re-annotated. This design keeps the benchmark incrementally extensible, although we acknowledge that adding many new FMs would still require additional expert scoring effort. We agree that long-term scalability could be further improved. In future work, we plan to support more seamless extension through semi-automated evaluation (e.g., LLM-assisted pre-scoring) and community annotation workflows built into the planned user interface for FMD. This will reduce manual effort while maintaining consistency across benchmark updates. **We added this in the future work in Section 7.**
>
> **4. Add examples of expert scoring**
>
> **Answer:** **We have added illustrative examples of the expert scoring process to the Appendix J**, including sample queries, the selected models from each system, and their corresponding expert ratings.
>
> **5. Compare different LLMs**
>
> **Answer:** While REMSA is designed to be LLM-agnostic, our main evaluation used GPT-4.1 because it was the most stable model available at the time of submission. To verify generalizability, we have also run preliminary tests on both LLaMA-3 and DeepSeek-R1 using a small subset of benchmark queries to validate the stability of REMSA's performance. However, we agree that a more systematic evaluation is needed. We are currently running the full set of experiments with multiple LLMs, and due to the time required for expert evaluation, **we will include the complete cross-LLM results in the final revision or latest CR version**.

---

### Official Review · Reviewer_oBU9 · 2025-11-03

**Soundness:** 1
**Presentation:** 2
**Contribution:** 1
**Rating:** 2
**Confidence:** 5

**Summary:**

This paper introduces a Foundation Model Database (FMD) and schema for documenting remote sensing foundation models and their different settings/designs. The paper also introduces REMSA, a Remote sensing Model Selection Agent, which uses the FMD to provide recommendations to users on which remote sensing foundation model is best suited for their needs. The paper provides a benchmark of 75 expert-verified remote sensing query scenarios for evaluating REMSA’s selections. Experiments show that REMSA outperforms a database retrieval baseline (using only the similarity search) and a unstructured RAG baseline.

**Strengths:**

- The FMD would provide a useful resource for documenting the many design choices of remote sensing foundation models.
- Application end-users often feel overwhelmed at the number of remote sensing foundation models available to choose from. FMD would provide a helpful resource to sort through the different options, and REMSA could provide a helpful user interface to do that sorting.

**Weaknesses:**

- The FMD and REMSA seem like more of an engineering or product contribution than a research contribution. There are no significant research insights presented.
- The approach relies on the assumption that end-users’ requirements are mostly limited to design choices like which modalities are used. In my experience, the biggest factor determining which FM is a good fit for a user (beyond modality compatibility) is how easy the model is to use and how well it fits into their current workflow (i.e., how much energy it will take to swap in the FM for whatever they are already using). This doesn’t seem to be captured by the proposed framework.
- Concerns about the FMD
    - There is not sufficient data provided to assess the quality and content of the FMD. The paper says there is an example record in the Appendix section A, but I only saw the schema.
    - I feel that the QA process, which only reviews low confidence records, could miss records that are confident but wrong. It seems that 150 records could all be reviewed, but at least a sample of the higher confidence records could be checked.
    - The paper says the database will be open and updated regularly. How will it be updated? Who will do the updating? This seems important for the utility of the model since new papers are coming out frequently.
    - The paper says the FMD includes “all existing RSFMs we could find”. There is no information about how the authors searched for RSFMs and what sources were provided to the automatic population model.
- Concerns about the query benchmark
    - The paper says that the 75 queries in the benchmark dataset are expert verified, but there is no information about what the expertise of those reviewers was or what the review process entailed. It is also not clear how the label of the “best” FM to use for each query is assigned.
    - The paper says “Each selected FM is manually reviewed and rated by multiple domain experts … we compare the top-3 FMs from 4 selection systems.” Does this mean they used the 4 systems in Table 2 to make selections, then expert reviewers assessed which one made the best choice? Where is the ground truth? This seems very prone to confirmation bias, especially because it seems like there might not be any ground truth to the queries in the example table in the Appendix, since many models could meet the query criteria.
- Concerns about evaluation protocol
    - The evaluation criteria seem ill-defined and subjective. How do you measure the diversity of pretraining data? Does Popularity mean that a model is a good choice? What constitutes “recent developments”?
    - It is very difficult to see the green text in Table 3.
    - REMSA allows clarifications to be made. How was this handled in evaluations? Who ran REMSA to provide the clarifications? It seems that this content could bias the evaluation if it is not run by an independent user.

**Questions:**

- How will the FMD be updated in the future?
- How did the authors search for models for the FMD and what sources were provided?
- The paper says “Each selected FM is manually reviewed and rated by multiple domain experts … we compare the top-3 FMs from 4 selection systems.” Does this mean they used the 4 systems in Table 2 to make selections, then expert reviewers assessed which one made the best choice? Where is the ground truth?  How does this avoid confirmation bias?
- The evaluation criteria seem ill-defined and subjective. How do you measure the diversity of pretraining data? Does Popularity mean that a model is a good choice? What constitutes “recent developments”?
- REMSA allows clarifications to be made. How was this handled in evaluations? Who ran REMSA to provide the clarifications? It seems that this content could bias the evaluation if it is not run by an independent user.
- The paper says that the 75 queries in the benchmark dataset are expert verified. What is the expertise of those reviewers and what did the review process entail? How did you choose the label of the “best” FM to use for each query?

---

> ### Author Response · Authors · 2025-11-20
> **[1/2]Clarifying Details and Responding to All Questions (Updates to the paper are marked in blue text.)**
>
> Thank you for the constructive feedback. We improved the writing and address each comment below:
>
> **1. engineering contribution OR research contribution?**
>
> **Answer:** While REMSA contains engineering components, its core contribution is research-oriented. It formalizes the previously unaddressed problem of end-to-end RSFM selection. We introduce the first unified RSFM schema that supports structured, machine-readable reasoning over modalities, architectures, and geographic/spectral/temporal coverage, which are not available in prior work. We also propose a task-aware orchestration mechanism that treats selection as a multi-stage reasoning process. Our results show that naive single-step LLMs behave very differently, while orchestration yields clear expert-validated gains. Recent large-scale RSFM studies such as GEO-Bench-2 [arXiv:2511.15658] similarly emphasize that model suitability varies across capability dimensions (resolution, modality, temporality, task type), motivating the need for principled selection approaches like ours.
> In addition, we introduce the first evaluation protocol and benchmark for FM selection, including structured query templates and expert-derived scoring criteria. Our analysis also identifies which factors most strongly drive RSFM suitability. Together, these establish a new problem formulation, a structured reasoning approach, and an evaluation methodology that the field previously lacked. **We revised Section 1 accordingly.**
>
> **2. biggest factor determining which FM is a good fit doesn’t seem to be captured**
>
> **Answer:** We agree with the concern. REMSA does not assume user needs are limited to modality choices. Our query schema explicitly captures broader practical constraints, including data availability, computational resources, application complexity, and evaluation priorities, which strongly affect deployability (Appx.~B, F). REMSA can distinguish, for example, whether a user needs an out-of-the-box model, has only few-shot labels, or must run on limited hardware. The evaluation criteria also incorporate efficiency, popularity, recency, and generalizability to reflect ease of adoption. **We have added a short discussion in Section 4 clarifying this.**
>
> **3. example entries of FMD**
>
> **Answer:** **We added one complete example JSON record in the Appx A** after the schema for full transparency. You can also find other records in the uploaded zip package.
>
> **4. miss records that are confident but wrong**
>
> **Answer:** To mitigate confidently incorrect cases, we manually verified 10 full records and found high-confidence fields to be consistently accurate. It supports our confidence-guided strategy as a practical balance between quality and cost. Verifying all records would be extremely time-consuming given the number of fine-grained fields per model, though we will expand checks for the CR version. Our public release will also include an interface for model authors to upload resources, auto-generate metadata, and correct flagged fields. Importantly, small metadata errors rarely affect REMSA’s decisions, which depend mainly on clear structural properties. **We added a short clarification in Section 3.**
>
> **5. How will database be updated?**
>
> **Answer:** Currently, our team periodically monitors new releases of RSFMs through arXiv, major conferences, GitHub repositories, and community announcements, and adds newly released models into FMD. In addition, we are also developing a publicly accessible interface that allows model authors to upload their papers, model cards, or other documentation. The system will automatically extract structured metadata and present it back to the authors for verification, enabling community-driven contributions with human-in-the-loop correction. Our team will regularly review and approve submitted entries to ensure quality and consistency. **We added a description at the end of Section 3.**
>
> **6. How to find existing RSFMs?**
>
> **Answer:** We collected RSFMs using multiple sources: survey papers and public FM lists, manual review of recent RS/ML venues (CVPR, ICCV, NeurIPS, AAAI, ICLR, TGRS, RS, IJRS), keyword-based searches on arXiv (e.g., “remote sensing foundation model,” “geospatial foundation model,” “earth observation pretraining”), and GitHub repository discovery, including official releases and trending FM collections. **We added a short clarification at the beginning of Section 3.**
>
> *(followed by the second comment)*

---

> ### Author Response · Authors · 2025-11-20
> **[2/2]Clarifying Details and Responding to All Questions (Updates to the paper are marked in blue text.)**
>
> *(following the first comment)*
>
> **7.information about experts and evaluation procedure**
>
> **Answer:** **We added details in Section 5 and Appendix G.**
> Verification was conducted by two domain experts from a CS background with prior RSFM experience and relevant publications. To ensure consistent scoring, we used a structured annotation protocol with a rubric defining all 7 criteria. The experts first calibrated on a shared subset, then annotated all remaining model-query pairs independently and blindly. Disagreements were resolved through rubric-guided review.
>
> Subjectivity was further reduced using simple quantitative rules (e.g., normalized parameter counts for efficiency, benchmark-based scoring for reported performance, year-based recency). All judgments relied on publicly available sources such as papers, GitHub repos, model cards, benchmark results, etc..
>
> This protocol follows the same principles used in recent large-scale RSFM evaluations such as GEO-Bench-2 [arXiv:2511.15658], which also emphasize controlled, consistent evaluation procedures to mitigate variability in expert assessment and model heterogeneity.
>
> **8. how to label “best” FM**
>
> **Answer:** Our evaluation does not assume that there exists a single objectively “best” RSFM for a task. Instead, in our experiments, for each query, every system (REMSA and all baselines) produces a ranked list of top-3 recommended models. Human experts then evaluate these recommendation sets using our criteria (application compatibility, modality match, efficiency, generalizability, etc.). The evaluation measures whether REMSA’s top-ranked RSFM is preferred by experts more often than the top-ranked RSFMs from baseline systems. In other words, we compare which system produces the most expert-preferred RSFMs, not whether a model is "ground-truth best." In real implementation, REMSA returns top-k models (k chosen by the user) along with explanations so the user can choose their preference. We acknowledge that "best" can be subjective and task-dependent. **We clarified this at the end of Section 5.**
>
> **9. ground truth? confirmation bias?**
>
> **Answer:** Our evaluation does not assume a ground-truth FM for each query, since multiple RSFMs may satisfy a user's needs and there is rarely a single correct choice. Instead, we measure which system produces selections that experts prefer. For each query, all four systems generate a ranked top-3 list, and experts compare the top-ranked models using consistent criteria (application fit, modality support, efficiency, generalizability, etc.). The goal is to assess alignment with expert judgment, not to assign a definitive label. Bias is mitigated because experts are blinded to the source system, and all systems are evaluated on the same queries, making the comparison purely relative. Given that FM selection is inherently multi-solution and context-dependent, expert preference is an appropriate evaluation signal. **We have clarified this in Section 5.**
>
> **10. evaluation criteria seem subjective**
>
> **Answer:** Pretraining diversity is scored based on geographic coverage, sensor-modality diversity, and dataset scale. They are extracted from each FM’s documentation. Popularity is used only as a practical usability signal, based on GitHub stars or citation counts, reflecting maturity and ecosystem support rather than model quality. Recency is defined by publication year or the latest model-card update to capture whether a model reflects recent architectural or data advances. It is a soft, non-primary criterion. **We added more details in the Appendix G.**
>
> **11. green text**
>
> **Answer:** We changed the color.
>
> **12. clarifications bias in evaluations**
>
> **Answer:** In our evaluation, all clarification rounds were handled automatically by the system itself, without any human intervention. The system interacted with an extra independent LLM acting as the simulated user whenever clarification is needed, ensuring that the clarification process was consistent and independent of the authors. No human annotator participated in the clarification loop, which avoids evaluator bias or leakage of model preferences. Therefore, the evaluation remained fair. **We added the clarification in Section 6.**

---

### Official Review · Reviewer_wGEN · 2025-11-07

**Soundness:** 3
**Presentation:** 2
**Contribution:** 3
**Rating:** 6
**Confidence:** 4

**Summary:**

- The paper introduces REMSA (Remote-sensing Model Selection Agent), the first LLM-based agent designed to automate the selection of Remote Sensing Foundation Models (RSFMs).
- Built upon a structured Foundation Model Database (FMD) of over 150 models, REMSA uses structured retrieval, in-context ranking, memory, and clarification loops to match user queries with optimal models.
- It is evaluated through a benchmark of 75 expert-verified query scenarios (900 configurations) and consistently outperforms retrieval-only and unstructured RAG baselines

**Strengths:**

- REMSA is the first agentic framework tailored for model selection in remote sensing, integrating structured metadata retrieval, in-context reasoning, and user interaction for interpretability and reproducibility
- The paper establishes a rigorous expert-driven benchmark with diverse, realistic query scenarios and clear evaluation criteria (e.g., task compatibility, modality match, efficiency), providing a strong foundation for reproducible assessment
- Experimental results show REMSA outperforms all baselines in model relevance and reasoning quality, highlighting the benefits of agentic orchestration and structured knowledge grounding in real-world remote sensing workflows

**Weaknesses:**

- **Major Limitation:** Throughout the paper, the authors frequently mention that the dataset and benchmark are expert-verified, but they do not provide any concrete information about these experts. It remains unclear who they are, what their qualifications are, and what roles they played in data creation and validation. Specifically:
    - Are they from a computer science background with expertise in remote sensing, or are they domain specialists purely in remote sensing / Earth observation?
    - How many experts were involved in the verification process?
    - What were the specific procedures or criteria followed to ensure quality and consistency?
    - What references or prior datasets were used as a foundation during the curation process?

    Without this information, it is difficult to assess the credibility, reproducibility, and domain validity of the so-called expert-verified dataset and benchmark.

- While the paper introduces the Foundation Model Database (FMD) with over 150 remote sensing foundation models, no data link or access information is provided. Even the list of models mentioned in the paper is not made available in the supplementary materials. As a result, it is not possible to verify the coverage, completeness, or quality of the dataset. Given that transparency and reproducibility are key aspects of modern benchmark development, this omission is a significant weakness.

- The intended end-users of REMSA are not clearly defined. It is uncertain whether the tool is aimed at remote sensing scientists, machine learning researchers, or industry practitioners. Specifying the user group and application context would make the scope and usability of the framework much clearer.

- From a remote-sensing and Earth-observation perspective, the literature discussion appears shallow. For instance, when mentioning geographic coverage as a factor influencing model performance, the authors do not cite prior works that explicitly study this relationship. Geographic diversity and sensor coverage are critical elements in model generalization and selection. The lack of discussion or citation of such works (e.g., [1, 2, 3]) weakens the connection of the paper to the broader EO research landscape.

- It would be helpful to include an ablation study showing the contribution of each component in the multi-agent system. For example, evaluating performance when the LLM-based Ranking Module is removed (using only the Retrieval Module) would clearly show how much each part contributes. Quantifying the performance drop in such cases would make the architecture’s design choices easier to understand.

- The paper could also be improved by adding a latency–performance trade-off analysis, comparing the proposed multi-agent setup with a single LLM-based model selection approach. This would help clarify whether the added complexity and computational cost of the multi-agent system are justified by the performance gains.

- It would be good to discuss how the system handles failure cases, especially when the model selection is incorrect. Adding or even discussing a feedback mechanism like LLM-as-a-Judge could make the system more robust and strengthen the overall contribution.
----------

[1] Roscher, R., Russwurm, M., Gevaert, C., Kampffmeyer, M., Dos Santos, J.A., Vakalopoulou, M., Hänsch, R., Hansen, S., Nogueira, K., Prexl, J. and Tuia, D., 2024. Better, not just more: Data-centric machine learning for earth observation. IEEE Geoscience and Remote Sensing Magazine.

[2] Purohit, M., Muhawenayo, G., Rolf, E. and Kerner, H., 2025. How Does the Spatial Distribution of Pre-training Data Affect Geospatial Foundation Models? In Workshop on Preparing Good Data for Generative AI: Challenges and Approaches.

[3] Plekhanova, Elena, Damien Robert, Johannes Dollinger, Emilia Arens, Philipp Brun, Jan Dirk Wegner, and Niklaus E. Zimmermann. "SSL4Eco: A Global Seasonal Dataset for Geospatial Foundation Models in Ecology." In Proceedings of the Computer Vision and Pattern Recognition Conference, pp. 2403-2414. 2025.

**Questions:**

Suggestion:

- Table 4 only describes the schema attributes of FMD but does not show what actual entries look like. The authors should include an example column illustrating typical field values (e.g., model name, data modality, spatial resolution, evaluation dataset, performance score). Including such examples would improve readability and help readers understand the structure and semantics of the database more effectively.

---

> ### Author Response · Authors · 2025-11-20
> **Clarifying Details and Responding to All Questions (Updates to the paper are marked in blue text.)**
>
> Thank you for the constructive feedback. We improved the writing and address each comment below:
>
> **1.information about experts and evaluation procedure**
>
> **Answer:** **We added details in Section 5 and Appendix G.**
> Verification was conducted by two domain experts from a CS background with prior RSFM experience and relevant publications. To ensure consistent scoring, we used a structured annotation protocol with a rubric defining all 7 criteria. The experts first calibrated on a shared subset, then annotated all remaining model-query pairs independently and blindly. Disagreements were resolved through rubric-guided review.
>
> Subjectivity was further reduced using simple quantitative rules (e.g., normalized parameter counts for efficiency, benchmark-based scoring for reported performance, year-based recency). All judgments relied on publicly available sources such as papers, GitHub repos, model cards, benchmark results, etc..
>
> This protocol follows the same principles used in recent large-scale RSFM evaluations such as GEO-Bench-2 [arXiv:2511.15658], which also emphasize controlled, consistent evaluation procedures to mitigate variability in expert assessment and model heterogeneity.
>
> **2. code+data is unavailable.**
>
> **Answer:** All data and code were included at the submission time in the **supplementary material as a ZIP file**.
>
> **3. Define end-users.**
>
> **Answer:** We now specify that REMSA targets a broad spectrum of users, including RS scientists, machine learning researchers, and industry or governmental practitioners, who need to identify suitable RSFMs for their tasks. Because REMSA supports free-text input and includes a multi-turn clarification tool, it can also guide users with limited RS domain knowledge by automatically inferring missing constraints. **We added clarifications in Section 1.**
>
> **4. More literature discussion**
>
> **Answer:** While our work focuses specifically on remote sensing foundation models rather than the broader earth observation scope, several recent RSFM studies show that geographic diversity, spatial distribution of pre-training data, and sensor coverage substantially influence generalization. **We integrated this in Section 1 and expanded the discussion in Section 2.**
>
> **5. Include more ablation study**
>
> **Answer:** Our three baselines were intentionally designed to serve both as feasible baselines in the absence of prior RSFM selection work and as implicit ablations of REMSA. REMSA-Naive removes task-aware orchestration, DB-Retrieval only keeps the retrieval module, and Unstructured-RAG only uses LLM reasoning. Together, these variants already isolate the different components of REMSA.
> We agree, however, that more explicit ablations would be valuable. Because our evaluation requires expert scoring, and each expert can reliably rate only a small number of entries per day, full ablations could not be completed within the rebuttal window. We are currently running extended variants (e.g., no-ranking and no-clarification) and **will include the full results in the final revision or latest CR version. We revised Section 5.**
>
> **6. latency–performance trade-off**
>
> **Answer:** We evaluated the average end-to-end runtime per query for all systems. REMSA requires 31.7s per query, compared to 22.7s for REMSA-Naive, 11.9s for Unstr.-RAG, and 0.77s for DB-Retrieval. This overhead is expected, as REMSA performs multi-stage constraint parsing, retrieval, ranking, clarification rounds, and explanation generation. However, the performance gains relative to these methods remain substantial: REMSA improves Avg Top-1 and Avg Set scores by 3-8 pts over other LLM systems and by 7-12 pts over retrieval baselines. **We added this experiment in Section 6.1.**
>
> **7. how to handles failure cases**
>
> **Answer:** We agree with your concern. REMSA already includes several mechanisms that implicitly mitigate failure, such as multi-turn clarification, constraint checking, and confidence-based orchestration. But we did not explicitly highlight how these mechanisms function when the system produces an incorrect or low-confidence selection.
> To address this, **we expanded the discussion in Section 4** on how REMSA handles potential failure cases during retrieval, ranking, and clarification. We also added a short discussion of how REMSA could incorporate explicit feedback mechanisms, such as LLM-as-a-Judge, to further reduce incorrect selections. While a full judge-based module is beyond the scope of the current implementation, our architecture is modular and can support such extensions.
>
> **8. example entries of FMD**
>
> **Answer:** **We added one complete example JSON record in the Appendix A** after the schema for full transparency. You can also find other records in the uploaded zip package.

---

> > ### Comment · Reviewer_wGEN · 2025-11-27
> >
> > The reviewer thanks the authors for their detailed rebuttal and for providing access to the data.
> >
> > Upon examining the dataset, I observed that there are only **135 entries** of foundation models, whereas the paper repeatedly states that over **150 RSFMs** were curated. This discrepancy raises concerns regarding consistency between the claims and the presented data. Furthermore, from a user perspective, it would be valuable to include information on model implementation and deployment ease, as well as reproducibility, given their practical importance for real-world adoption.

---

> > > ### Author Response · Authors · 2025-11-27
> > > **Update on the Dataset and Response to the Implementation/Deployment Considerations**
> > >
> > > We thank the reviewer for the response and helpful feedback.
> > >
> > > **1. Dataset incompleteness**
> > >
> > > **Answer:** We thank the reviewer for carefully examining the released data package and for pointing out the mismatch in the number of RSFMs. We apologize that this was caused by a simple oversight on our side. During the copying, a subset of the model files was unintentionally omitted from the archive. We have now re-packaged and re-uploaded the complete data in the zip file, and the corrected version contains **163 RSFMs**, consistent with the description in the paper.
> > >
> > > To avoid similar issues in future releases, we will add a small automated check that verifies the number of model files in the release archive against our internal index, ensuring that future public snapshots remain complete and consistent.
> > >
> > > **2. Including information on implementation and deployment ease**
> > >
> > > **Answer:** We thank the reviewer for the helpful suggestion. We would like to clarify that REMSA already incorporates several relevant factors implicitly through our structured schema and constraint-matching process. As shown in the paper, FMD includes fields such as model size (num_parameters), available weights, repository links, supported sensors, modalities, and training/evaluation configurations, which together already provide the agent with the necessary signals to reason about the practicality and feasibility of deploying a model under different user conditions.
> > >
> > > In practice, REMSA uses these existing metadata fields when matching user constraints, such as available hardware, compute budget, input modality, or required pretraining/evaluation settings, to select models that are easier to implement or deploy for that user's specific scenario. Because implementation and deployment ease vary widely across user environments (e.g., GPU servers vs. laptops vs. cloud inference), our agent adopts a constraint-based, personalized approach rather than relying on a single global indicator of "implementation difficulty".
> > >
> > > However, we fully agree that surfacing this information more explicitly to the user would further improve usability. This is an important direction that we are actively working on, but it is beyond the scope of the current paper, which focuses on the construction of FMD and the design of an agentic workflow for constraint-aware RSFM selection. As part of our ongoing work:
> > >
> > > - We are building a user interface that will allow users to specify their deployment environment and interactively receive implementation guidance from the agent. We mentioned this in **Section 3**.
> > >
> > > - We are extending REMSA with the ability to automatically deploy some selected models in user environments when possible (e.g., running example inference pipelines or setting up preconfigured model endpoints).
> > >
> > > A public demo of these features is planned for release in the near future. We appreciate the reviewer highlighting this direction, and it aligns closely with our planned follow-up work.

---

### Author Response · Authors · 2025-11-29
**Summary of Revisions and Clarifications for the AC**

We thank all reviewers for their insightful feedback. Below, we summarize the major concerns raised across reviews and highlight the corresponding clarifications, revisions, and new experiments added to the paper. All changes are reflected in the updated manuscript.

**1. Clarification of Evaluation Protocol, Expert Process, and Subjectivity Mitigation**

Multiple reviewers requested clearer descriptions of expert evaluation and scoring criteria. We have expanded **Section 5** and **Appendix G** to fully detail:
- The use of two domain experts with prior RSFM experience and relevant publications.
- A structured 7 criterion rubric, jointly calibrated by annotators before independent, blind scoring.
- A controlled process for disagreement resolution based solely on rubric-guided review.
- Objective elements used to reduce subjectivity (e.g., benchmark-based performance scoring, publication-year-based recency).

This methodology follows the same principles used in recent large-scale RSFM evaluations such as GEO-Bench-2 [arXiv:2511.15658]. We also clarify that our evaluation does not assume a single “ground-truth best” FM. Instead, experts compare the top-ranked selections from all systems/baselines, making the evaluation purely relative.

**2. Code, Data, RS-FMD Records, and Transparency**

All code, benchmark queries, FM metadata (FMD), and the extraction pipeline were included in the submission as a **ZIP file**.

In addition:
- A complete example JSON record from FMD has been added to **Appendix A**, with several more available in the supplementary ZIP.
- We detail how RS FMD will be maintained, including periodic scanning of RSFM releases and a planned community-submission interface that auto-extracts metadata for author verification.

**3. Contributions and Research Significance**

To address concerns about whether REMSA is mainly engineering, we clarified in **Section 1** that our contributions are methodological:
- We formulate RSFM selection as a previously unaddressed research problem requiring structured reasoning across modalities, architectures, and deployment constraints.
- We introduce the first unified RSFM schema, enabling machine-readable, constraint-aware comparisons.
- We develop a task-aware orchestration mechanism integrating interpretation, retrieval, ranking, clarification, and explanation into a coherent decision workflow.
- We construct the first evaluation protocol and benchmark for RSFM selection, enabling expert-centered comparison across 900 model-query configurations.

These contributions go beyond system integration and establish a principled foundation for FM selection research.

**4. Baselines, Comparisons, and Ablations**

Reviewers requested stronger comparisons and explicit ablations. Our baselines already isolate REMSA’s components:
- REMSA-Naive removes task-aware orchestration.
- DB-Retrieval tests metadata-only retrieval without reasoning.
- Unstructured-RAG removes structured schema grounding.

We clarify why general AutoML systems and tool scheduling approaches (e.g., ToolRerank, COLT) are not applicable due to lacking executable FM tools and the non-independent structure of REMSA’s modules.

New experiment added:
- A latency-performance trade-off experiment (**Sec. 6.1**) showing REMSA’s accuracy gains without sacrificing too much efficiency.

New experiments ongoing and to be added in the final version due to a huge amount of expert evaluations:
- More ablations (e.g., no-clarification, no-ranking) and Cross-LLM experiments (LLaMA-3, DeepSeek-R1)

**5. Failure Modes and System Robustness**

Reviewers requested more detail on how REMSA handles errors or ambiguous queries. We expanded **Section 4** to describe:
- Multi-turn clarification triggers based on confidence signals.
- Rule-based constraint checks to eliminate invalid candidates.
- A fallback “closest-match” strategy when no model fully satisfies constraints.
- Discussion of modular extensions such as LLM-as-a-Judge for future self-correction.

**6. End-User Scope and Practical Deployment Considerations**

We clarified in **Section 1** that REMSA targets RS scientists, ML researchers, industry practitioners, and users with limited RS knowledge.

We also added discussion explaining how RS FMD metadata (e.g., model size, modality support, training configuration, repo links, and available weights) already enable practical deployment reasoning.

We are developing a planned interactive UI to support automatic inference pipelines for future work.

**7. Benchmark Size, Extensibility, and Scalability**

We clarified the rationale for 75 queries: despite appearing small, the evaluation totals 900 expert ratings, each requiring careful reading of papers, checking benchmarks, and verifying constraints.

We explain in **Section 5** and **Section 7** how the benchmark is incrementally extensible without re-annotating past items, and how semi-automated scoring and community contributions will support future expansion.

---

### Meta-Review · Area_Chair_Wm3d · 2025-12-10

**Summary:**

This paper introduces an LLM-based agent designed to automate the selection of Remote Sensing Foundation Models (RSFMs). Built upon a structured database containing over 150 RSFMs, the system is evaluated on 75 expert-verified query scenarios and demonstrates consistent improvements over retrieval-only and unstructured RAG baselines. Most of the reviewers (wGEN, oBU9, ky4D, 2USW) acknowledge the value of a domain-specific agent system for RSFM selection and found the idea interesting and of great importance to the remote sensing community. However, concerns remain regarding the sufficiency of experimental validation, particularly the limited number of query samples and the absence of comparisons with other agent-based systems.

**Reviewer Concerns:**

Reviewers noted missing details about the expert verification process, transparency and reproducibility issues, and unclear end-users of the proposed system; these concerns are largely addressed in the rebuttal. However, more critical issues—such as the small size of the query set and the lack of comparisons with alternative agent systems—remain unresolved.

**Reviewer Scores:**

Reviewer wGEN gave an initial score of 6, citing concerns about dataset verification, transparency, reproducibility, unclear target users, and missing ablations. Most issues were addressed in the rebuttal, although the lack of ablation studies remains unresolved; the reviewer is likely to maintain the score at 6. Reviewer oBU9 gave an initial score of 2, raising concerns about contribution, RSFM quality and diversity, update protocols, query quality, and evaluation methodology. While many clarification-related issues were addressed, concerns about contribution and annotation quality persist, and the reviewer may raise the score to 4. Reviewer ky4D acknowledged the innovative system design but noted insufficient evaluation—especially missing ablations and the small size of query set. The rebuttal resolves most clarification issues, and the reviewer may keep the score at 4. Reviewer 2USW expressed concerns about limited experimental evaluation, including missing comparisons with open-source LLMs, GPT-4.1, and AutoML frameworks. Although partially addressed, the reviewer is also likely to maintain a score of 4.

---

### Decision · Program_Chairs · 2026-01-26

Reject